# Unconstrained Dynamic Regret via Sparse Coding

**Zhiyu Zhang**[*]
Harvard University
zhiyuz@seas.harvard.edu

**Ashok Cutkosky**
Boston University
ashok@cutkosky.com

**Ioannis Ch. Paschalidis**
Boston University
yannisp@bu.edu

## Abstract

Motivated by the challenge of nonstationarity in sequential decision making, we study Online Convex Optimization (OCO) under the coupling of two problem structures: the domain is unbounded, and the comparator sequence $u_1, \ldots, u_T$ is arbitrarily time-varying. As no algorithm can guarantee low regret simultaneously against all comparator sequences, handling this setting requires moving from minimax optimality to comparator adaptivity. That is, sensible regret bounds should depend on certain complexity measures of the comparator relative to one's prior knowledge. This paper achieves a new type of such adaptive regret bounds leveraging a sparse coding framework. The complexity of the comparator is measured by its energy and its sparsity on a user-specified dictionary, which offers considerable versatility. For example, equipped with a wavelet dictionary, our framework improves the state-of-the-art bound [JC22] by adapting to both ($i$) the magnitude of the comparator average $\|\bar{u}\| = \|\sum_{t=1}^{T} u_t/T\|$, rather than the maximum $\max_t \|u_t\|$; and ($ii$) the comparator variability $\sum_{t=1}^{T} \|u_t - \bar{u}\|$, rather than the uncentered sum $\sum_{t=1}^{T} \|u_t\|$. Furthermore, our analysis is simpler due to decoupling function approximation from regret minimization.

## 1 Introduction

Nonstationarity is prevalent in sequential decision making, which poses a critical challenge to the vast majority of existing approaches developed offline. Consider weather forecasting for example [SBG+21]. A meteorologist typically starts from the governing physical equations and simulates them online using high performance computing; the imperfection of this physical model can lead to time-varying patterns in its forecasting error. Alternatively, a machine learning scientist may build a data-driven model from historical weather datasets, but its online deployment is subject to distribution shifts. If the structure of such nonstationarity can be exploited in our algorithm, then we may expect better forecasting performance. This paper investigates the problem from a theoretical angle – we aim to improve nonstationary online decision making by incorporating *temporal representations*.

Concretely, we study *Online Convex Optimization* (OCO), which is a repeated game between us (the player) and an adversarial environment $\mathcal{E}$. In each (the $t$-th) round, with a mutually known Lipschitz constant $G$:

1. We make a prediction $x_t \in \mathbb{R}^d$ based on the observations before the $t$-th round.
2. The environment $\mathcal{E}$ reveals a convex loss function $l_t : \mathbb{R}^d \to \mathbb{R}$ dependent on our prediction history $x_1, \ldots, x_t$; $l_t$ is $G$-Lipschitz with respect to $\|\cdot\|_2$.
3. We suffer the loss $l_t(x_t)$.

The game ends after $T$ rounds, and then, our total loss is compared to that of an alternative sequence of predictions $u_1, \ldots, u_T \in \mathbb{R}^d$. Without knowing the time horizon $T$, the environment $\mathcal{E}$ and the

---

[*]Work done at Boston University. Future versions available at https://arxiv.org/abs/2301.13349.

37th Conference on Neural Information Processing Systems (NeurIPS 2023).

*comparator sequence* $\{u_t\}_{t \in \mathbb{Z}}$, our goal is to achieve low *unconstrained dynamic regret*

$$\text{Regret}_T(u_{1:T}) := \sup_{\mathcal{E}} \left[ \sum_{t=1}^{T} l_t(x_t) - \sum_{t=1}^{T} l_t(u_t) \right]. \tag{1}$$

Fixing any comparator $\{u_t\}_{t \in \mathbb{Z}}$: if such an expression can be upper-bounded by a sublinear function of $T$, then asymptotically, in any environment, we perform at least as well as the $\{u_t\}_{t \in \mathbb{Z}}$ sequence.

The above setting deviates from the most standard setting of OCO [Haz16, Ora19] in two ways.

- **Structure 1.** The domain $\mathbb{R}^d$ is unbounded.
- **Structure 2.** The comparator is allowed to be time-varying.

While the latter has been studied extensively in the literature (since [Zin03]) to account for nonstationarity, most existing approaches require a *time-invariant bounded domain* to set the hyperparameters properly, which, to some extent, limits the amount of nonstationarity they can handle. One might argue that most practical problems have a finite range, which could be heuristically estimated from offline datasets. However, such a heuristic is not robust in nature, as underestimates will invalidate the theoretical analysis, and overestimates will make the regret bound excessively conservative. It is thus important to study the more challenging unconstrained dynamic setting[2] combining the two problem structures, where algorithms cannot rely on pre-selected range estimates at all.

Taking a closer look at their analysis, it is perhaps a little surprising that these two problem structures share a common theme, despite being studied mostly separately. In either the unconstrained static setting [MO14, OP16, CO18] or the bounded dynamic setting [Zin03, HW15, ZLZ18], the standard form of *minimax optimality* [ABRT08, AABR09, RS14b] becomes vacuous, as it is impossible to guarantee that $\sup_{u_{1:T}} \text{Regret}_T(u_{1:T})$ is sublinear in $T$. Circumventing this issue relies on *comparator adaptivity*[3] – instead of only depending on $T$, any appropriate regret upper bound, denoted by $\text{Bound}_T(u_{1:T})$, should also depend on the comparator $u_{1:T}$ through a certain *complexity measure*. Intuitively, despite the intractability of hard comparators, nonvacuous bounds can be established against "easy ones". A total loss bound then follows from the *oracle inequality*

$$\sum_{t=1}^{T} l_t(x_t) \le \inf_{u_{1:T}} \left[ \sum_{t=1}^{T} l_t(u_t) + \text{Bound}_T(u_{1:T}) \right]. \tag{2}$$

A crucial observation is that the complexity of $u_{1:T}$ is not uniquely defined: one could imagine bounding $\text{Regret}_T(u_{1:T})$ by many different non-comparable functions of $u_{1:T}$. Essentially, this complexity measure serves as a *Bayesian prior*:[4] choosing it amounts to assigning different priorities to different comparators $u_{1:T} \in \mathbb{R}^{d \times T}$. The associated algorithm guarantees lower $\text{Bound}_T(u_{1:T})$ against comparators with higher priority, and due to Eq.(2), the total loss of our algorithm is low if some of these high priority comparators *actually* achieve low loss $\sum_{t=1}^{T} l_t(u_t)$. Such a Bayesian reasoning highlights the importance of versatility in this workflow: in order to place an arbitrary application-dependent prior, we need a versatile algorithmic framework that adapts to a wide range of complexity measures. This leads to the limitations of existing results, discussed next.

To our knowledge, [JC22] is the only existing work that considers our setting. Two unconstrained dynamic regret bounds are presented based on three statistics of the comparator sequence, the *maximum range* $M := \max_t \|u_t\|_2$, the *norm sum* $S := \sum_{t=1}^{T} \|u_t\|_2$ and the *path length* $P := \sum_{t=1}^{T-1} \|u_{t+1} - u_t\|_2$. First, using a 1D unconstrained static algorithm as a simple range scaler, the paper achieves [JC22, Lemma 10]

$$\text{Regret}_T(u_{1:T}) \le \tilde{O}\left( \sqrt{(M+P)MT} \right). \tag{3}$$

---

[2]It is known that unconstrained OCO algorithms can also handle *time-varying* (but not necessarily bounded) domains in a black-box manner [Cut20, Section 4].

[3]In general, adaptivity means achieving near minimax optimality simultaneously for many restricted subclasses of the problem, where minimax optimality is well-defined [Joh19, Chapter 6].

[4]The prior can be selected on the fly, depending on the observation history. This brings key practical benefits: Appendix E discusses how an empirical forecaster based on domain knowledge or deep learning could be "robustified" using our framework.

Table 1: Comparison in almost static environments. Each row improves the previous row (omitting logarithmic factors), c.f., Appendix A.

| Algorithm | $P$-dependent bound | $K$-switching regret | Example 1 | Example 2 |
|---|---|---|---|---|
| ADER [ZLZ18] (meta-expert OGD) | $\tilde{O}\left(\sqrt{(D+P)DT}\right)$ | $\tilde{O}\left(D\sqrt{(1+K)T}\right)$ | N/A | $\tilde{O}(T^{3/4})$ |
| [JC22, Algorithm 6] (range scaling) | $\tilde{O}\left(\sqrt{(M+P)MT}\right)$ | $\tilde{O}\left(M\sqrt{(1+K)T}\right)$ | $\tilde{O}(T)$ | $\tilde{O}(T^{3/4})$ |
| [JC22, Algorithm 2] (centered MD) | $\tilde{O}\left(\sqrt{(M+P)S}\right)$ | $\tilde{O}\left(\sqrt{(1+K)MS}\right)$ | $\tilde{O}(T^{3/4})$ | $\tilde{O}(T^{3/4})$ |
| Ours (Haar OLR) | $\tilde{O}\left(\|\bar{u}\|_2\sqrt{T}+\sqrt{P\bar{S}}\right)$ | $\tilde{O}\left(\|\bar{u}\|_2\sqrt{T}+\sqrt{K\bar{E}}\right)$ | $\tilde{O}(\sqrt{T})$ | $\tilde{O}(\sqrt{T})$ |

Then, by developing a customized mirror descent approach, most of the effort is devoted to improving $MT$ to $S$ [JC22, Theorem 4], i.e., adapting to the magnitude of individual $u_t$.

$$\text{Regret}_T(u_{1:T}) \leq \tilde{O}\left(\sqrt{(M+P)S}\right). \tag{4}$$

Despite the strengths of these results and their nontrivial analysis, a shared limitation is that both bounds depend explicitly on the path length $P$. Intuitively, it means that good performance is only guaranteed in *almost static* environments: in the typical situation of $S = \Theta(T)$, these bounds are only sublinear when $P = o(T)$, which rules out important persistent dynamics such as periodicity. Moreover, even the second bound still depends on $MS$ instead of a finer characterization of each individual $u_t$'s magnitude. That is, the mission of removing $M$ is not fully accomplished yet.[5]

The goal of this paper is to extend comparator adaptivity to a wider range of complexity measures. For almost static environments in particular, quantitative benefits will be obtained from specific instances of this general approach.

## 1.1 Contribution

The contributions of this paper are twofold.

1. First, we present an algorithmic framework achieving a new type of unconstrained dynamic regret bounds. It is based on a conversion to vector-output *Online Linear Regression* (OLR): given a dictionary $\mathcal{H}$ of orthogonal feature vectors spanning the *sequence space* $\mathbb{R}^{dT}$, we use an unconstrained static OCO algorithm to linearly aggregate these feature vectors, which are themselves time-varying prediction sequences. Such a procedure guarantees

$$\text{Regret}_T(u_{1:T}) \leq \tilde{O}\left(\sqrt{E \cdot \text{Sparsity}_{\mathcal{H}}}\right), \tag{5}$$

where $E = \sum_{t=1}^T \|u_t\|_2^2$ is the *energy* of the comparator $u_{1:T}$, and $\text{Sparsity}_{\mathcal{H}}$ measures the sparsity of $u_{1:T}$ on the dictionary $\mathcal{H}$.[6] Both $E$ and $\text{Sparsity}_{\mathcal{H}}$ are unknown beforehand.

   Compared to [JC22], the main advantage of this framework is its versatility. Prior knowledge on the *transform domain* can be incorporated by picking $\mathcal{H}$, and favorable algorithmic properties can be conveniently inherited from static online learning.

2. Our second contribution is quantitative: although [JC22] is specifically crafted to handle almost static environments, we show that equipped with a *Haar wavelet* dictionary, our framework actually guarantees better bounds (Table 1) in this setting, which is a surprising finding to us.

   - With the *comparator average* $\bar{u} := \sum_{t=1}^T u_t/T$ and the *first order variability* $\bar{S} := \sum_{t=1}^T \|u_t - \bar{u}\|_2$, our Haar wavelet algorithm guarantees

$$\text{Regret}_T(u_{1:T}) \leq \tilde{O}\left(\|\bar{u}\|_2\sqrt{T}+\sqrt{P\bar{S}}\right).$$

---

[5]The significance of this issue could be seen through an analogy to (static $D$-bounded domain) *gradient adaptive* OCO: although there are algorithms achieving the already adaptive $O\left(D\sqrt{G\sum_{t=1}^T \|g_t\|_2}\right)$ static regret bound, the hallmark of gradient adaptivity is the so-called "second-order bound" $O\left(D\sqrt{\sum_{t=1}^T \|g_t\|_2^2}\right)$, popularized by ADAGRAD [DHS11]. In a rough but related sense, we aim to achieve "second order comparator adaptivity", which is only manifested in the less studied dynamic regret setting.

[6]For conciseness, we omit $u_{1:T}$ in the notation. Throughout this paper, the regularity parameters on the RHS of the regret bound generally depend on $u_{1:T}$. A list of these parameters is presented in Appendix A, including their relations.

It improves Eq.(4) by $(i)$ a better dependence on the comparator magnitude ($\sqrt{MS} \to \|\bar{u}\|_2 \sqrt{T}$); and $(ii)$ decoupling the bias $\bar{u}$ from the characterization of variability ($\sqrt{PS} \to \sqrt{P\bar{S}}$).

- With the *number of switches* $K := \sum_{t=1}^{T-1} \mathbf{1}[u_{t+1} \neq u_t]$ and the *second order variability* $\bar{E} := \sum_{t=1}^{T} \|u_t - \bar{u}\|_2^2$, the same Haar wavelet algorithm guarantees an *unconstrained switching regret bound*

$$\text{Regret}_T(u_{1:T}) \leq \tilde{O}\left(\|\bar{u}\|_2 \sqrt{T} + \sqrt{K\bar{E}}\right),$$

which improves the existing $\tilde{O}\left(\sqrt{(1+K)MS}\right)$ bound resulting from Eq.(4) and $P = O(KM)$.

Due to the *local property* of wavelets, our algorithm runs in $O(d \log T)$ time per round, matching that of the baselines. As for the regret, our bounds are never worse than the baselines, and in two examples corresponding to $\|\bar{u}\|_2 \ll M$ and $\bar{S} \ll S$, they reduce to clearly improved rates in $T$. Furthermore, our analysis follows from the generic regret bound Eq.(5) and the *wavelet approximation theory*, providing an intriguing connection between disparate fields.

The paper concludes with an application in fine-tuning time series forecasters, where unconstrained dynamic OCO is naturally motivated. Due to limited space, this is deferred to Appendix E, with experiments that support our theoretical results.

## 1.2 Related work

Our paper addresses the connection between unconstrained OCO and dynamic OCO. Although they both embody the idea of comparator adaptivity, unified studies have been scarce.

**Unconstrained OCO** To obtain static regret bounds in OCO, *Online Gradient Descent* (OGD) [Zin03] is often the default approach. With learning rate $\eta$, it guarantees $O(\eta^{-1}\|u\|_2^2 + \eta T)$ regret with respect to any *unconstrained* static comparator $u \in \mathbb{R}^d$, and the optimal choice in hindsight is $\eta = O(\|u\|_2 / \sqrt{T})$. Without the oracle knowledge of $\|u\|_2$, it is impossible to tune $\eta$ optimally. To address this issue, a series of works (also called *parameter-free online learning*) [SM12, MA13, GY14, MO14, OP16, FKMS17, CO18, FRS18, MK20, CLW21, ZCP22] developed vastly different strategies to achieve the *oracle optimal rate* $O(\|u\|\sqrt{T})$ up to logarithmic factors. That is, the algorithm performs as if the complexity measure $\|u\|$ is known beforehand.

There is certain flexibility in the choice of the norm $\|\cdot\|$: $L_1$ and $L_2$ norm bounds were presented in [SM12], while Banach norm bounds were developed by [FRS18, CO18]. Historically, the connection between the $L_1$ norm and sparsity has powered breakthroughs in batch data science, including LASSO [Tib96] and compressed sensing [CRT06]. However, the parallel path in online learning remains less studied: while the sparsity implication of the $L_1$ norm adaptive bounds has been discussed in the literature [SM12, Ger13, vdH19], there is in general a lack of downstream investigations with concrete benefits. In this paper, we show that the sparsity of the comparator can be naturally associated to the *structural simplicity* of a nonstationary environment.

**Dynamic OCO** Comparing against dynamic sequences is a classical research topic. It is clear that one cannot go beyond linear regret in the worst case, therefore various notions of complexity should be introduced.

- The closest topic to ours is the *universal dynamic regret*, where the regret bound adapts to the complexity of an arbitrary $u_{1:T}$ on a bounded domain with $L_p$-diameter $D$. In the most common framework, the complexity measure is an $L_{p,q}$ norm of the difference sequence $\{u_{t+1} - u_t\}$, such as the $L_{p,1}$ norm, i.e., the path length $P = \sum_{t=1}^{T-1} \|u_{t+1} - u_t\|_p$ [HW01].[7] Omitting the dependence on the dimension $d$ (thus also the choice of $p$), the optimal bound under convex Lipschitz losses is $\tilde{O}\left(\sqrt{(D+P)DT}\right)$ [Zin03, HW15, JRSS15, GS16, ZLZ18], while the accelerated rate[8] $\tilde{O}(P^{2/3}T^{1/3} \vee 1)$ can be achieved with strong convexity [BW21, BW22]. Improvements have been

---

[7]Motivated by nonparametric statistics, the $L_{p,q}$ norm $\left(\sum_{t=1}^{T-1} \|u_{t+1} - u_t\|_p^q\right)^{1/q}$ with $q > 1$ is associated to more homogeneous measures of comparator smoothness [BW19, BW21].

[8]Further omitting the dependence on the diameter $D$.

studied under the additional smoothness assumption [ZZZZ20, ZZZZ21]. These bounds subsume results in *switching (a.k.a., shifting) regret*, where the complexity of $u_{1:T}$ is measured by its number of switches $K$, as $P$ is dominated by $DK$.

A notable exception is the *dynamic model* framework from [HW15, ZLZ18]. Still considering a bounded domain, it takes a collection of dynamic models as input, which are mappings from the domain to itself. Then, the complexity of a comparator $u_{1:T}$ is measured by how well it can be reconstructed by the best dynamic model in hindsight. Essentially, the use of temporal representations is somewhat similar to the dictionary in our framework. The important difference is that instead of using the best feature (or the best convex combination of the features) to measure the comparator, we use *linear combinations* of the features – this allows handling unconstrained domains through subspace modeling.

- Besides the universal dynamic regret, there are other notions of dynamic regret that do not induce oracle inequalities like Eq.(2), including (*i*) the *restricted dynamic regret* [YZJY16, ZYY$^+$17, BW19, BW20, BZW21], which depends on the complexity of certain *offline optimal* comparators;[9] and (*ii*) regret bounds that depend on the *functional variation* $\sum_{t=1}^{T-1} \max_x |l_t(x) - l_{t+1}(x)|$ [BGZ15, CWW19]. They are not as relevant to our purpose, due to being vacuous on unbounded domains under the linear losses – this is an important setting in our investigation.

**Unconstrained (universal) dynamic regret**   To our knowledge, [JC22] is the only work studying the universal dynamic regret without a bounded domain, whose contributions have been summarized in our Introduction. Here we survey some negative results in the literature.

- The restricted dynamic regret is a special case of the universal dynamic regret, therefore lower bounds for the former apply to the latter as well. For convex Lipschitz losses [YZJY16] and strongly convex losses [BW19], any algorithm should suffer the dynamic regret of $\Omega(P)$ and $\Omega(P^2)$, respectively.

- For dynamic OCO on bounded domains, a recurring analysis goes through the notion of *strong adaptivity* [DGSS15]: one first achieves low *static* regret bounds on *every subinterval* of the time horizon $[1:T]$, and then assembles these local bounds appropriately to bound the global dynamic regret [ZYZ$^+$18, Cut20, BW21, BW22]. Following this route in the unconstrained setting appears to be challenging, as [JC22, Section 4] showed that (a natural form of) strong adaptivity cannot be achieved there.

Additional discussions of related works are deferred to Appendix B, including the more general problem of *online nonparametric regression*, the more specific problem of *parametric time series forecasting*, and other orthogonal uses of sparsity in online learning.

## 1.3   Notation

For two integers $a \leq b$, $[a:b]$ is the set of all integers $c$ such that $a \leq c \leq b$. The brackets are removed when on the subscript, denoting a tuple with indices in $[a:b]$. Treating all vectors as column vectors, $\mathrm{span}(A)$ represents the column space of a matrix $A$. $\log$ is natural logarithm when the base is omitted, and $\log_+(\cdot) := 0 \vee \log(\cdot)$. polylog denotes a poly-logarithmic function of its input. $0$ represents a zero vector whose dimension depends on the context.

## 2   The general sparse coding framework

This section presents our sparse coding framework, achieving the generic sparsity adaptive regret bound Eq.(5). The key idea is to view online learning on the sequence space $\mathbb{R}^{dT}$, rather than the default domain $\mathbb{R}^d$. Despite its central role in signal processing (e.g., the *Fourier transform*), such a view is (in our opinion) under-explored by the online learning community.[10]

---

[9]Notably, [BW19, BW20] creatively employed wavelet techniques to detect change-points of the environment, which, to the best of our knowledge, is the only existing use of wavelets in the online learning literature.

[10]Possibly due to the emphasis on the static regret: the sequence $u_{1:T}$ collapses into a time-invariant $u$, which is contained in $\mathbb{R}^d$.

## 2.1 Setting

To begin with, we follow the conventions in online learning [Haz16, Ora19] to linearize convex losses. Consider that instead of the full loss function $l_t$, we only observe its subgradient $g_t \in \partial l_t(x_t)$ at our prediction $x_t$. By using the linear loss $\langle g_t, \cdot \rangle$ as a surrogate, we can still upper bound the regret Eq.(1) due to $l_t(x_t) - l_t(u) \leq \langle g_t, x_t - u \rangle$. The linear loss problem is also called *Online Linear Optimization* (OLO), where each observation $g_t$ is a $d$ dimensional vector satisfying $\|g_t\|_2 \leq G$.

Now, consider the length $T$ sequences of predictions $x_{1:T}$, gradients $g_{1:T}$ and comparators $u_{1:T}$. Let us flatten everything and treat them as $dT$ dimensional vectors, concatenating per-round quantities in $\mathbb{R}^d$. These are called *signals*. The comparator statistics could be more succinctly represented using vector notations, e.g., the energy $E = \sum_{t=1}^{T} \|u_t\|_2^2 = \|u_{1:T}\|_2^2$.

Our framework requires a *dictionary* matrix $\mathcal{H} \in \mathbb{R}^{dT \times N}$, possibly revealed online, whose columns are $N$ nonzero *feature vectors*. We write $\mathcal{H}$ in an equivalent block form as $[h_{t,n}]_{1 \leq t \leq T, 1 \leq n \leq N}$, where each block $h_{t,n} \in \mathbb{R}^{d \times 1}$. The accompanied linear transform $u = \mathcal{H}\hat{u}$ relates a signal $u \in \mathbb{R}^{dT}$ to a coefficient vector $\hat{u} \in \mathbb{R}^N$ (if it exists). Adopting the convention in signal processing, we will call $\mathbb{R}^{dT}$ the *time domain*, and $\mathbb{R}^N$ the *transform domain*. In general, symbols without hat refer to time domain quantities, while their transform domain counterparts are denoted with hat.

Summarizing the above, we consider the following concise interaction protocol.[11] Despite its parametric appearance, our main focus is on the *nonparametric* regime, where the dictionary size $N$ scales with the amount of data $T$.

**Vector-output OLR with linear losses**  In the $t$-th round, our algorithm observes a $d$-by-N feature matrix $\mathcal{H}_t := [h_{t,n}]_{1 \leq n \leq N}$, linearly combines its columns into a prediction $x_t \in \mathbb{R}^d$, receives a loss gradient $g_t \in \mathbb{R}^d$, and then suffers the linear loss $\langle g_t, x_t \rangle$. We assume that[12] $\|h_{t,n}\|_2 \leq 1$, $\sum_{t=1}^{T} \|h_{t,n}\|_2^2 \geq 1$ and $\|g_t\|_2 \leq G$. The performance metric is the unconstrained dynamic regret defined in Eq.(1).

## 2.2 Main result

In a nutshell, our strategy is to apply an unconstrained static OLO algorithm on the transform domain, and in a coordinate-wise fashion. This is remarkably simple, but also contains a few twists. To make it concrete, let us start with a single feature vector.

**Size 1 dictionary**  Consider an index $n \in [1 : N]$, which is associated to the feature $h_{1:T,n} := [h_{1,n}, \ldots, h_{T,n}] \in \mathbb{R}^{dT}$. We suppress the index $n$ and write it as $h_{1:T} = [h_1, \ldots, h_T]$. For any comparator $u_{1:T} \in \text{span}(h_{1:T})$, there exists $\hat{u} \in \mathbb{R}$ such that $u_{1:T} = h_{1:T}\hat{u}$. The cumulative loss of $u_{1:T}$ can be rewritten as

$$\langle g_{1:T}, u_{1:T} \rangle = \langle g_{1:T}, h_{1:T} \rangle \, \hat{u} = \sum_{t=1}^{T} \langle g_t, h_t \rangle \, \hat{u},$$

which is the loss of the coefficient $\hat{u}$ in a 1D OLO problem with surrogate loss gradients $\langle g_t, h_t \rangle$. Essentially, to compete with a 1D comparator subspace $\text{span}(h_{1:T})$, it suffices to run a 1D static regret algorithm $\mathcal{A}$ that competes with $\hat{u} \in \mathbb{R}$. Such a procedure is presented as Algorithm 1.

It still remains to choose the static algorithm $\mathcal{A}$. Technically, all known static comparator adaptive algorithms can be applied. As an illustrative example, we adopt the FREEGRAD algorithm [MK20], which simultaneously achieves static comparator adaptivity and *second order gradient adaptivity* [DHS11].[13] Its pseudocode and static regret bound are presented in Appendix C.1 for completeness.

In summary, our single feature learner (Algorithm 1) has the following simplified guarantee, with the full gradient adaptive version deferred to Appendix C.

---

[11]Despite also using features, the considered setting slightly differs from the standard notion of regression, as the loss function here does not necessarily have a minimizer. We use the term OLR for cleaner exposition.

[12]The assumptions on the features are mild: an important special case is $\max_t \|h_{t,n}\|_2 = 1$, as in the Haar wavelet dictionary. We impose these assumptions to apply unconstrained static algorithms verbatim.

[13]A gradient adaptive regret bound refines our definition Eq.(1) by depending on the *actually encountered* environment $\mathcal{E}$ as well. FREEGRAD enjoys another favorable property called *scale-freeness*: the predictions are invariant to any positive scaling of the loss gradients and the Lipschitz constant $G$.

---

**Algorithm 1** Sparse coding with size 1 dictionary.

---

**Require:** An algorithm $\mathcal{A}$ for static 1D unconstrained OLO with $G$-Lipschitz losses; and a nonzero feature vector $h_{1:T} \subset \mathbb{R}^{dT}$.

1: **for** $t = 1, 2, \ldots,$ **do**
2:     Receive $h_t \in \mathbb{R}^d$.
3:     If $h_t$ is nonzero, query $\mathcal{A}$ for its output, and assign it to $\hat{x}_t \in \mathbb{R}$; otherwise, $\hat{x}_t$ is arbitrary.
4:     Predict $x_t = \hat{x}_t h_t \in \mathbb{R}^d$, and receive the loss gradient $g_t \in \mathbb{R}^d$.
5:     If $h_t$ is nonzero, compute $\hat{g}_t = \langle g_t, h_t \rangle$ and send it to $\mathcal{A}$ as its surrogate loss gradient.
6: **end for**

---

**Lemma 2.1.** *Let $\varepsilon > 0$ be an arbitrary hyperparameter for* FREEGRAD *(Algorithm 3 in Appendix C.1). Applying its 1D version as the static subroutine, for all $T \in \mathbb{N}_+$ and $u_{1:T} \in \mathrm{span}(h_{1:T})$, Algorithm 1 guarantees*

$$\mathrm{Regret}_T(u_{1:T}) \leq \varepsilon G + \|u_{1:T}\|_2 G \cdot \mathrm{polylog}\left(\max_t \|u_t\|_2, T, \varepsilon^{-1}\right).$$

Note that the hyperparameter $\varepsilon$ can be arbitrarily small. Further neglecting poly-logarithmic factors, the bound is essentially $\tilde{O}(G\|u_{1:T}\|_2)$.

**General dictionary** Given the above single feature learner, let us turn to the general setting with $N$ features. We run $N$ copies of Algorithm 1 in parallel, aggregate their predictions, and the regret bound sums Lemma 2.1, similar to [Cut19] in the static setting. The pseudocode is presented as Algorithm 2, and the regret bound is Theorem 1.

---

**Algorithm 2** Sparse coding with general dictionary.

---

**Require:** A dictionary $\mathcal{H} = [h_{t,n}]$, where $h_{t,n} \in \mathbb{R}^d$; and a hyperparameter $\varepsilon > 0$.

1: For all $n \in [1 : N]$, initialize a copy of Algorithm 1 as $\mathcal{A}_n$. It runs the 1D version of Algorithm 3 as a subroutine, with hyperparameter $\varepsilon/N$.
2: **for** $t = 1, 2, \ldots,$ **do**
3:     Receive $\mathcal{H}_t = [h_{t,n}]_{1 \leq n \leq N}$. For all $n$, send $h_{t,n}$ to $\mathcal{A}_n$, and query its prediction $w_{t,n}$.
4:     Predict $x_t = \sum_{n=1}^{N} w_{t,n}$.
5:     Receive loss gradient $g_t$, and send it to $\mathcal{A}_1, \ldots, \mathcal{A}_N$ as loss gradients.
6: **end for**

---

**Theorem 1.** *Consider any collection of signals $z^{(n)} \in \mathrm{span}(h_{1:T,n})$, $\forall n$. We define its reconstruction error (for the comparator $u_{1:T}$) as $z^{(0)} = u_{1:T} - \sum_{n=1}^{N} z^{(n)} \in \mathbb{R}^{dT}$. Then, for all $T \in \mathbb{N}_+$ and $u_{1:T} \in \mathbb{R}^{dT}$, Algorithm 2 guarantees*

$$\mathrm{Regret}_T(u_{1:T}) \leq \varepsilon G + G\left(\sum_{n=1}^{N} \left\|z^{(n)}\right\|_2\right) \cdot \mathrm{polylog}\left(\max_{t,n}\left\|z_t^{(n)}\right\|_2, T, N, \varepsilon^{-1}\right) + G\sum_{t=1}^{T}\left\|z_t^{(0)}\right\|_2,$$

*where $z_t^{(n)} \in \mathbb{R}^d$ is the $t$-th round component of the sequence $z^{(n)} \in \mathbb{R}^{dT}$.*

To interpret this very general result, let us consider a few concrete settings.

- **Static regret.** If the size $N = d$ and the dictionary $\mathcal{H}_t = I_d$, then for any static comparator ($u_t = u \in \mathbb{R}^d$), we can let $z^{(n)}$ be the projection of the sequence $u_{1:T}$ onto $\mathrm{span}(h_{1:T,n})$. This leaves zero reconstruction error, i.e., $u_{1:T} = \sum_{n=1}^{N} z^{(n)}$. Theorem 1 reduces to

$$\mathrm{Regret}_T(u_{1:T}) \leq \varepsilon G + \|u\|_1 G\sqrt{T} \cdot \mathrm{polylog}\left(\|u\|_\infty, T, d, \varepsilon^{-1}\right), \tag{6}$$

  which recovers a standard $\tilde{O}(\|u\|_1 \sqrt{T})$ bound in coordinate-wise unconstrained static OLO [Ora19, Section 9.3]. The gradient adaptive version yields a better $\tilde{O}(\|u\|_2 \sqrt{T})$ bound, c.f., Appendix C.2.

- **Orthogonal dictionary.** Entering the dynamic realm, we now consider the situation where feature vectors are orthogonal (standard in signal processing), and the comparator $u_{1:T} \in \mathrm{span}(\mathcal{H})$. Same as the static setting, we are free to define $z^{(n)}$ as the projection

$$z^{(n)} = \langle h_{1:T,n}, u_{1:T} \rangle \frac{h_{1:T,n}}{\|h_{1:T,n}\|_2^2}.$$

Due to orthogonality, the projection preserves the energy of the time domain signal, i.e, $E = \|u_{1:T}\|_2^2 = \sum_{n=1}^N \|z^{(n)}\|_2^2$. By further defining $\text{Sparsity}_{\mathcal{H}} := (\sum_{n=1}^N \|z^{(n)}\|_2)^2 / \sum_{n=1}^N \|z^{(n)}\|_2^2$ (arbitrary when the denominator is zero), Theorem 1 reduces to

$$\text{Regret}_T(u_{1:T}) \leq \tilde{O}\left(\sqrt{E \cdot \text{Sparsity}_{\mathcal{H}}}\right). \tag{7}$$

Note that as the squared $L_1/L_2$ ratio, $\text{Sparsity}_{\mathcal{H}}$ is a classical sparsity measure [HR09] of the decomposed signals $\{z^{(n)}\}_{1 \leq n \leq N}$: if there are only $N_0 \leq N$ nonzero vectors within this collection, then $\text{Sparsity}_{\mathcal{H}} \leq N_0$ due to the Cauchy-Schwarz inequality. Therefore, the generic sparsity adaptive bound Eq.(7) depends on $(i)$ the energy of the comparator $u_{1:T}$; and $(ii)$ the sparsity of its representation, without knowing either condition beforehand. The easier the comparator is (low energy, and sparse on $\mathcal{H}$), the lower the bound becomes.

- **Overparameterization.** So far we have only considered $N \leq dT$, where feature vectors can be orthogonal. However, a key idea in signal processing is to use redundant features ($N \gg dT$) to obtain sparser representations. Theorem 1 implies a *feature selection* property in this context: since it applies to *any* decomposition of $u_{1:T}$, as long as $u_{1:T}$ can be represented by a subset $\tilde{\mathcal{H}}$ of orthogonal features within $\mathcal{H}$, the regret bound adapts to $\text{Sparsity}_{\tilde{\mathcal{H}}}$, the sparsity of $u_{1:T}$ measured on $\tilde{\mathcal{H}}$. That is, we are theoretically justified to assemble smaller dictionaries into a larger one – the regret bound adapts to the quality of the optimal (comparator-dependent) sub-dictionary $\tilde{\mathcal{H}}$.

How to choose the dictionary $\mathcal{H}$? In practice, we may use prior knowledge on the dynamics of the environment. For example, if the environment is periodic, such as the weather or the traffic, then a good choice could be the Fourier dictionary. Similarly, wavelet dictionaries are useful for piecewise regular environments. Another possibility is to learn the dictionary from offline datasets, which is also called *representation learning*. Overall, such prior knowledge is not required to be *correct* – our algorithm can take any dictionary as input, and the regret bound naturally adapts to its quality. The established connection between adaptivity and signal structures is a key benefit of our framework.

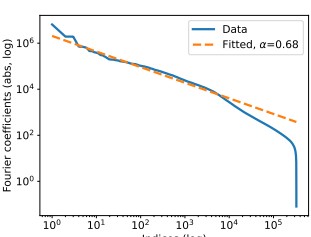

Figure 1: The power law.

**Power law**  For a more specific discussion, let us consider an empirically justified setup. In signal processing, the study of sparsity has been partially motivated by the *power law* [Pri21]: under the standard Fourier or wavelet transforms, the $n$-th largest transform domain coefficient of many real signals can have magnitude roughly proportional to $n^{-\alpha}$, where $\alpha \in (0.5, 1)$. We also observe this phenomenon from a weather dataset, with details presented in Appendix E.1. Figure 1 plots the sorted Fourier coefficients of an actual temperature sequence, on a log-log scale. A fitted dashed line is shown in orange, with (negative) slope $\alpha = 0.68$.

When the power law holds, our bound Eq.(7) has a more interpretable form. Assuming $d = 1$ and $N = T$,

$$\text{Sparsity}_{\mathcal{H}} = \frac{(\sum_{n=1}^T n^{-\alpha})^2}{\sum_{n=1}^T n^{-2\alpha}} = O\left(T^{2-2\alpha}\right).$$

In a typical setting of $E = \Theta(T)$, we obtain a sublinear $\tilde{O}(T^{1.5-\alpha})$ dynamic regret bound.

## 3 The Haar OLR algorithm

This section presents the quantitative contributions of this paper: despite its generality, our sparse coding framework can improve existing results [JC22]. Our workhorse is the ability of wavelet bases to sparsely represent smooth signals.

### 3.1 Haar wavelet

Wavelet is a fundamental topic in signal processing, with long lasting impact throughout modern data science. Roughly, the motivation is that a signal can simultaneously exhibit nonstationarity at different time scales, such as slow drifts and fast jumps, therefore to faithfully represent it, we should apply feature vectors with different resolutions. We will only use the simplest Haar wavelets, which is already sufficient. Readers are referred to [Mal08, Joh19] for a thorough introduction to this topic.

Specifically, we start from the 1D setting ($d = 1$) with a dyadic horizon ($T = 2^m$, for some $m \in \mathbb{N}_+$). The Haar wavelet dictionary consists of $T$ (unnormalized) orthogonal feature vectors, indexed by a *scale* parameter $j \in [1 : \log_2 T]$ and a *location* parameter $l \in [1 : 2^{-j}T]$. Given a $(j, l)$ pair, define a feature $h^{(j,l)} = [h_1^{(j,l)}, \ldots, h_T^{(j,l)}] \in \mathbb{R}^T$ entry-wise as

$$h_t^{(j,l)} = \begin{cases} 1, & t \in [2^j(l-1) + 1 : 2^j(l-1) + 2^{j-1}]; \\ -1, & t \in [2^j(l-1) + 2^{j-1} + 1 : 2^j l]; \\ 0, & \text{else.} \end{cases}$$

It means that $h^{(j,l)}$ is only nonzero on a length-$2^j$ interval, while changing its sign once in the middle of this interval. Collecting all the $(j, l)$ pairs yield $T - 1$ features; then, we incorporate an extra all-one feature $h^* = [1, \ldots, 1]$ to complete this size $T$ dictionary.

The defined features can be assembled into the columns of a matrix $\mathrm{Haar}_m$. To help with the intuition, $\mathrm{Haar}_2$ with $T = 4$ is presented in Eq.(8). The columns from the left to the right are $h^*$, $h^{(2,1)}$, $h^{(1,1)}$ and $h^{(1,2)}$. Observe that they are orthogonal, and the norm assumption from Section 2.1 is satisfied. Therefore, our sparsity adaptive regret bound Eq.(7) is applicable.

$$\mathrm{Haar}_2 = \begin{bmatrix} 1 & 1 & 1 & 0 \\ 1 & 1 & -1 & 0 \\ 1 & -1 & 0 & 1 \\ 1 & -1 & 0 & -1 \end{bmatrix}. \tag{8}$$

Given this 1D Haar wavelet dictionary, we apply a minor variant of Algorithm 2 to prevent the dimension $d$ from appearing in the regret bound. When $d = 1$, the algorithm is exactly Algorithm 2, where intuitions are most clearly demonstrated. Then, the doubling trick [SS11, Section 2.3.1] is adopted to relax the knowledge of $T$. The pseudocode is presented as Algorithm 5 in Appendix D.

**Computation**  An appealing property is that most Haar wavelet features are supported on short local intervals. Despite $N = T$, there are only $\log_2 T$ active features in each round. Therefore, the runtime of our algorithm is $O(d \log T)$ per round, matching that of all the baselines we compare to. This local property holds for compactly supported wavelets, most notably the *Daubechies family* [Dau88, CDV93]. The latter can represent more general, piecewise polynomial signals.

### 3.2  Main result

For almost static environments, our Haar OLR algorithm guarantees the following bounds, by relating comparator smoothness to the sparsity of its Haar wavelet representation. Different from [JC22] which only contains $P$-dependent bounds, we also provide a $K$-switching regret bound, in order to avoid using $P = O(KM)$.[14] Interestingly, the proofs of the following two bounds are quite different: the first uses *exact sparsity*, while the second uses *approximate sparsity*.

**Theorem 2** (Switching regret). *For all $T \in \mathbb{N}_+$ and $u_{1:T} \in \mathbb{R}^{dT}$, Algorithm 5 guarantees*

$$\mathrm{Regret}_T(u_{1:T}) \leq \tilde{O}\left( \|\bar{u}\|_2 \sqrt{T} + \sqrt{K\bar{E}} \right). \tag{9}$$

**Theorem 3** (Path length bound). *For all $T \in \mathbb{N}_+$ and $u_{1:T} \in \mathbb{R}^{dT}$, Algorithm 5 guarantees*

$$\mathrm{Regret}_T(u_{1:T}) \leq \tilde{O}\left( \|\bar{u}\|_2 \sqrt{T} + \sqrt{P\bar{S}} \right). \tag{10}$$

It can be verified (Appendix A) that for *all* comparators $u_{1:T}$, our bounds are at least as good as prior works (Table 1). The optimality is a more subtle issue, as one should compare *upper bound functions* (of $u_{1:T}$) to *lower bound functions* in a global manner, rather than comparing the exponents of $T$ in minimax online learning.

Nonetheless, we present two examples of $u_{1:T}$, where the improvement can be clearly seen through better exponents of $T$. To give it a concrete background, suppose we want to sequentially predict a 1D time series $z_1, \ldots, z_T \in \mathbb{R}$. This could be formulated as a OCO problem where the decision $x_t$ there is our prediction of $z_t$, and the loss function is the absolute loss $l_t(x) = |x - z_t|$. A natural

---

[14]Recall that one of our motivations is to remove $M$ from the existing bounds.

choice of the comparator is the ground truth sequence $z_{1:T}$, and due to Eq.(2), any upper bound on $\mathrm{Regret}_T(z_{1:T})$ also upper-bounds the total forecasting loss of our algorithm. Below we present specific 1D comparator sequences $u_{1:T}$ to demonstrate the strength of our results, which could be intuitively thought as the true time series $z_{1:T}$ in this more restricted discussion.

**Example 1** (Tracking outliers)**.** *Consider the situation where $u_{1:T}$ has a locally outlying scale: we set all the instantaneous comparators $u_t$ to 1, except $k \leq \sqrt{T}$ consecutive members which are set to $\sqrt{T}$. Crucially, $|\bar{u}| = O(1)$ and $\bar{S} = O(k\sqrt{T})$, while $M = \sqrt{T}$ and $S = \Theta(T)$. With details deferred to Appendix D.7, both our bounds, i.e., Eq.(9) and (10), are $\tilde{O}(\sqrt{kT})$, while the fine baseline Eq.(4) is $\tilde{O}(T^{3/4})$, and the coarse baseline Eq.(3) is $\tilde{O}(T)$. The largest gain is observed when $k$ is a constant, i.e., the comparator is subject to a short but large perturbation.*

**Example 2** (Persistent oscillation)**.** *Consider the situation where $\bar{u} = 1$, and all the instantaneous comparators oscillate around $\bar{u}$: $u_t = \bar{u} + \alpha_t/\sqrt{T}$. $\alpha_t = 1$ or $-1$, and it only switches sign for $k$ times. Notice that $\bar{S} = \sqrt{T}$, while $S = \Theta(T)$. All the baselines are $\tilde{O}\left(\sqrt{T} + k^{1/2}T^{1/4}\right)$, while both our bounds are $\tilde{O}(\sqrt{T})$. The largest gain is observed when $k = T - 1$, i.e., the comparator switches in every round.*

In summary, we show that existing bounds are suboptimal, while the optimality of our results remains to be studied. It highlights the importance of *comparator energy* and *variability* in the pursuit of better algorithms, which have not received enough attention in the literature. Next, we briefly sketch the proofs of these bounds.

**Proof sketch** The switching regret bound mostly follows from a very simple observation: if a sequence is constant throughout the support of a Haar wavelet feature, then its transform domain coefficient for this feature is zero. As features on the same scale $j$ do not overlap, a $K$-switching comparator can only induce $K$ nonzero coefficients on the $j$-th scale. There are at most $K \log_2 T$ nonzero coefficients in total, therefore $\mathrm{Sparsity}_{\mathcal{H}} = \tilde{O}(K)$. The bound Eq.(9) is obtained by applying this argument after taking out the average of $u_{1:T}$.

As for the path length bound, the idea is to consider the *reconstructed* sequences, using transform domain coefficients on a single scale $j$. These are usually called *detail sequences* in the wavelet literature [Mal08]. Each detail sequence has a relatively simple structure, whose path length and variability can be associated to the magnitude of its transform domain coefficients. Moreover, as these detail sequences are certain "locally averaged" and "globally centered" versions of the actual comparator $u_{1:T}$, their regularities are dominated by the regularity of $u_{1:T}$ itself. In combination, this yields a relation between $P\bar{S}$ and the coefficients' $L_1$ norm, i.e., $\sum_{n=1}^{N} \left\| z^{(n)} \right\|_2$ in Theorem 1, from which the bound is established.

Compared to the analysis of [JC22], the key advantage of our analysis is the decoupling of function approximation from the generic sparsity-based regret bound. The former is algorithm-independent, while the latter can be conveniently combined with advances in static online learning. With the help of approximation theory (e.g., Fourier features, wavelets, and possibly deep learning further down the line), intuitions are arguably clearer in this way, and solutions could be more precise (compared to analyses that "mix" function approximation with regret minimization).

**Additional discussion** Finally, due to limited space, we defer additional discussion of our technical results to Appendix F, including

- The related use of *Multi-Resolution Analysis* (MRA) in the existing online learning literature.
- The comparison between Lipschitz and strongly convex losses in unconstrained dynamic OCO.

## 4   Conclusion

This paper presents a unified study of unconstrained and dynamic online learning, where the two problem structures are naturally connected via comparator adaptivity. Building on the synergy between static parameter-free algorithms and temporal representations, we develop an algorithmic framework achieving a generic sparsity-adaptive regret bound. Equipped with the wavelet dictionary, our framework improves the quantitative results from [JC22], by adapting to finer characterizations of the comparator sequence.

## Acknowledgments and Disclosure of Funding

We thank Vivek Goyal for helpful pointers to the signal processing literature, and the NeurIPS reviewers for their constructive feedback. This research was partially supported by the NSF under grants CCF-2200052, DMS-1664644, and IIS-1914792, by the ONR under grant N00014-19-1-2571, by the DOE under grant DE-AC02-05CH11231, by the NIH under grant UL54 TR004130, and by Boston University.

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

# Appendix

**Organization** Appendix A summarizes a list of comparator statistics involved in our theoretical analysis. Appendix B surveys additional related works. Appendix C, D and E respectively present details of our general sparse coding framework, its special version with wavelet dictionaries, and applications in time series forecasting. Finally, Appendix F presents additional discussion of our technical results.

## A List of comparator statistics

A major task in comparator adaptive online learning is finding suitable statistics to quantify the regularity of a comparator sequence. Several such statistics are defined throughout this paper, which are summarized in Table 2. Note that for the definition of sparsity on the last line, we assume the dictionary $\mathcal{H}$ is orthogonal, and the sequence $u_{1:T}$ is contained in its span. Then, $z^{(n)}$ is defined as the projection of $u_{1:T}$ onto a feature vector $h_{1:T,n}$, i.e.,

$$z^{(n)} = \langle h_{1:T,n}, u_{1:T} \rangle \frac{h_{1:T,n}}{\|h_{1:T,n}\|_2^2}.$$

This is well defined due to our assumption $\|h_{1:T,n}\|_2 \geq 1$ from Section 2.1.

| Name | Notation | Definition |
|---|---|---|
| Maximum range | $M$ | $\max_t \|u_t\|$ |
| Comparator average | $\bar{u}$ | $\frac{1}{T} \sum_{t=1}^T u_t$ |
| Path length | $P$ | $\sum_{t=1}^{T-1} \|u_{t+1} - u_t\|_2$ |
| Norm sum | $S$ | $\sum_{t=1}^T \|u_t\|_2$ |
| First order variability | $\bar{S}$ | $\sum_{t=1}^T \|u_t - \bar{u}\|_2$ |
| Energy | $E$ | $\sum_{t=1}^T \|u_t\|_2^2$ |
| Second order variability | $\bar{E}$ | $\sum_{t=1}^T \|u_t - \bar{u}\|_2^2$ |
| Number of switches | $K$ | $\sum_{t=1}^{T-1} \mathbf{1}[u_{t+1} \neq u_t]$ |
| Sparsity on a dictionary $\mathcal{H}$ | $\text{Sparsity}_{\mathcal{H}}$ | $\frac{(\sum_{n=1}^N \|z^{(n)}\|_2)^2}{\sum_{n=1}^N \|z^{(n)}\|_2^2}$ |

Table 2: A list of comparator statistics.

Next, let us discuss their relations, in order to interpret our quantitative contribution more clearly (Table 1). It is clear that $S \leq MT$, therefore the fine baseline Eq.(4) from [JC22] improves the coarse one Eq.(3); comparing their associated switching regret bounds follows the same reasoning.

To compare our results to the baselines, observe that

$$\|\bar{u}\|_2 \sqrt{T} = \left\| \frac{1}{\sqrt{T}} \sum_{t=1}^T u_t \right\|_2 \leq \sum_{t=1}^T \frac{1}{\sqrt{T}} \|u_t\|_2 \leq \sqrt{E} \leq \sqrt{MS},$$

$$\bar{S} = \sum_{t=1}^T \|u_t - \bar{u}\|_2 \leq \sum_{t=1}^T \|u_t\|_2 + T\|\bar{u}\|_2 = \sum_{t=1}^T \|u_t\|_2 + \left\| \sum_{t=1}^T u_t \right\|_2 \leq 2S,$$

$$\bar{E} = \sum_{t=1}^T \|u_t - \bar{u}\|_2^2 \leq 2M\bar{S} \leq 4MS.$$

Therefore,

$$\|\bar{u}\|_2 \sqrt{T} + \sqrt{P\bar{S}} \leq \sqrt{MS} + 2\sqrt{2PS} = O\left(\sqrt{(M+P)S}\right),$$

$$\|\bar{u}\|_2 \sqrt{T} + \sqrt{K\bar{E}} \leq \sqrt{MS} + 2\sqrt{KMS} = O\left(\sqrt{(1+K)MS}\right).$$

That is, both our path-length-dependent bound and the switching regret bound are at least as good as the results from [JC22]. Concrete benefits are demonstrated in Example 1 and 2.

Finally, as a sanity check, our path-length-dependent bound does not violate the lower bound $\Omega(P)$: even when $\bar{u} = 0$,

$$P = \sum_{t=1}^{T-1} \|u_{t+1} - u_t\|_2 \leq \sum_{t=1}^{T-1} \|u_{t+1} - \bar{u}\|_2 + \sum_{t=1}^{T-1} \|u_t - \bar{u}\|_2 \leq 2\bar{S},$$

therefore our bound is never better than $\tilde{O}(P)$.

# B   More on related work

**Online regression**   Our sparse coding framework converts unconstrained dynamic OCO to a special form of online regression. The standard setting of the latter [RS14a] considers a repeated game as well: in each round, we observe a covariate $x_t \in \mathbb{R}^d$, make a prediction $\hat{y}_t \in \mathbb{R}$ (which depends on $x_t$), and then observe a label $y_t \in \mathbb{R}$. The performance metric is the minimax regret under the square loss

$$\text{Regret}_T(\mathcal{F}) = \sum_{t=1}^{T} (\hat{y}_t - y_t)^2 - \inf_{f \in \mathcal{F}} \sum_{t=1}^{T} (f(x_t) - y_t)^2.$$

Roughly, the problem is of a nonparametric type if the complexity of the function class $\mathcal{F}$ is not fixed a priori, but grows with $T$ (i.e., the amount of data).

Overall, such an online regression problem is highly general, as static OCO is recovered if $x_t$ is time-invariant. The setting we utilize is a variant with ($i$) vector output; ($ii$) general convex losses; ($iii$) $x_t$ specified by the dictionary, possibly being sparse itself (e.g., wavelets); and ($iv$) the function class $\mathcal{F}$ being linear, but unbounded. As discussed in Footnote 11, our setting deviates from the conventional definition of regression, as a general convex loss function does not necessarily have minimizers. We adopt the terminology of "regression" for streamlined exposition.

Existing works on online nonparametric regression [RS14a, GG15] have established the relation of this problem to certain path length characterizations of dynamic regret. However, the generality of this setting makes the analysis challenging, and especially, algorithms can be computationally expensive. With a bounded domain assumption (on predictions $\hat{y}_t$), a recent breakthrough [BW21] simultaneously achieved several notions of optimality for path-length-dependent bounds, with efficient computation. Readers are referred to [BW21, Appendix A] for a thorough discussion of this line of works.

For the special case of *Online Linear Regression* (OLR) with square losses, the celebrated VAW forecaster [AW01, Vov01] guarantees $O(N \log T)$ regret against any unbounded coefficient vector $\hat{u} \in \mathbb{R}^N$, where $N$ is the dimension of the feature space. Such a fast rate becomes vacuous in the nonparametric regime (when $N > T$) [GY14], therefore [Ger13] proposed a *sparsity regret bound* $\tilde{O}(\|\hat{u}\|_0)$ and an accompanying inefficient algorithm as its high dimensional generalization. Efficient computation was addressed by [GW18], but the obtained result only applies to bounded $\hat{u}$. In a rough sense, such sparsity regret bounds are the square loss and feature-based analogue of the $L_1$-norm parameter-free bounds in OLO [Ora19, Chapter 9]. They are also closely related to *sparsity oracle inequalities* in statistics, as reviewed by [Ger13].

**Parametric time series models**   For time series forecasting, most prior works are devoted to parametric strategies with strong inductive bias, such as the ARMA model, state space models, and more recent deep learning models. Online learning has been applied to such models as well [AHMS13, AHZ15, AM16, KM16, HLS$^+$18], leading to forecasting guarantees under mild statistical assumptions. When convexity is present, some of these problems could be reframed as special cases of our OLR problem, with a constant-size dictionary that does not grow with $T$; for example, learning the *autoregressive* model corresponds to defining the features as the fixed-length observation history. Also, Appendix E shows that given a parametric time series forecaster (possibly without performance guarantees), our algorithm can be applied on top of it, in order to provably correct its nonstationary bias.

**Other sparsity topics in OL**   Finally, we review other sparsity-related topics in online learning, which do not fit into the scope of this paper. [LLZ09, Xia09, DSSST10, SST11] considered using

online learning to solve batch $L_1$ regularized problems. The goal is to achieve sparse predictions instead of sparsity adaptive regret bounds. [Kal14, FKK16, KKLP17] studied *online sparse regression*, where only a subset of features are available in each round. The challenge is to handle bandit feedback in OLR.

## C   Detail on the general framework

This section presents details on our general sparse coding framework. Appendix C.1 introduces the static subroutine we adopt from [MK20]. Appendix C.2 proves our main results, but with additional gradient adaptivity compared to the main paper.

### C.1   Unconstrained static subroutine

The following static OCO algorithm and its guarantee are due to [MK20, Section 3.1]. We assume that $\|\hat{g}_t\|_2 \leq \hat{G}$, and $\hat{G} > 0$.

---

**Algorithm 3** FREEGRAD [MK20, Definition 4]: scale-free and gradient adaptive unconstrained static OLO.

---

**Require:** A hyperparameter $\varepsilon > 0$; dimension $d$; Lipschitz constant $\hat{G}$.
1: Initialize a gradient sum counter $s = 0 \in \mathbb{R}^d$ and a variance counter $v = \hat{G}^2$.
2: **for** $t = 1, 2, \ldots$ **do**
3:    Predict
$$\hat{x}_t = -\varepsilon s \cdot \frac{(2v + \hat{G}\|s\|_2)\hat{G}^2}{2(v + \hat{G}\|s\|_2)^2\sqrt{v}} \cdot \exp\left(\frac{\|s\|_2^2}{2v + 2\hat{G}\|s\|_2}\right).$$
4:    Observe the loss gradient $\hat{g}_t$.
5:    Update $s \leftarrow s + \hat{g}_t$, and $v \leftarrow v + \hat{g}_t^2$.
6: **end for**

---

**Lemma C.1** (Theorem 20 of [MK20])**.** *With any hyperparameter $\varepsilon > 0$, for all $T \in \mathbb{N}_+$ and $\hat{u} \in \mathbb{R}$, Algorithm 3 guarantees*

$$\sum_{t=1}^{T} \langle \hat{g}_t, \hat{x}_t - \hat{u} \rangle \leq \varepsilon\hat{G} + \left[2\|\hat{u}\|_2\sqrt{V_T \log_+\left(\frac{2\|\hat{u}\|_2 V_T}{\varepsilon\hat{G}^2}\right)}\right] \vee \left[4\|\hat{u}\|_2 \hat{G}\log\left(\frac{4\|\hat{u}\|_2\sqrt{V_T}}{\varepsilon\hat{G}}\right)\right],$$

*where*

$$V_T = \hat{G}^2 + \sum_{t=1}^{T}\|\hat{g}_t\|_2^2.$$

### C.2   Proof of the main result

We now present the analysis of our general sparse coding framework. The following lemma is a slightly more general version of Lemma 2.1 in the main paper, which characterizes the performance of our single direction learner (Algorithm 1). Recall that $g_t \in \partial l_t(x_t)$ from the OCO-OLO reduction.

**Lemma C.2** (Lemma 2.1, full)**.** *Let $\varepsilon > 0$ be an arbitrary hyperparameter for Algorithm 3. Applying its 1D version as the static subroutine, for all $T \in \mathbb{N}_+$ and $u_{1:T} \in \mathrm{span}(h_{1:T})$, against any adversary $\mathcal{E}$, Algorithm 1 guarantees*

$$\sum_{t=1}^{T} l_t(x_t) - \sum_{t=1}^{T} l_t(u_t) \leq \varepsilon G + \left(G\frac{\|u_{1:T}\|_2}{\|h_{1:T}\|_2} + \sqrt{\sum_{t=1}^{T} \langle g_t, u_t \rangle^2}\right) \cdot \mathrm{polylog}\left(\max_t \|u_t\|_2, T, \varepsilon^{-1}\right).$$

The simplified form (Lemma 2.1) is recovered by using $\|h_{1:T}\|_2 \geq 1$ and $\|g_t\|_2 \leq G$.

*Proof of Lemma 2.1.* Subsuming poly-logarithmic factors, the static regret bound of our static subroutine (Algorithm 3) can be written as

$$\sum_{t=1}^{T} \hat{g}_t \left( \hat{x}_t - \hat{u} \right) \leq \varepsilon \hat{G} + |\hat{u}| \left( \hat{G} + \sqrt{\sum_{t=1}^{T} \hat{g}_t^2} \right) \cdot \text{polylog} \left( |\hat{u}|, T, \varepsilon^{-1} \right),$$

where $\hat{u}$ is any 1D static comparator that the subroutine handles.

Now, for any single-directional comparator $u_{1:T} \in \text{span}(h_{1:T})$ considered in this lemma, there exists $\hat{u} \in \mathbb{R}$ such that $u_{1:T} = \hat{u} h_{1:T}$. The dynamic regret can be rewritten as

$$\sum_{t=1}^{T} l_t(x_t) - \sum_{t=1}^{T} l_t(u_t) \leq \sum_{t=1}^{T} \langle g_t, x_t - u_t \rangle = \sum_{t=1}^{T} \langle g_t, h_t \hat{x}_t - h_t \hat{u} \rangle = \sum_{t=1}^{T} \hat{g}_t (\hat{x}_t - \hat{u}),$$

and the RHS can be bounded using the static regret bound above. Note that $|\hat{g}_t| = |\langle g_t, h_t \rangle| \leq G$, therefore the surrogate Lipschitz constant $\hat{G}$ from the static regret bound can be assigned to $G$.

In summary,

$$\sum_{t=1}^{T} l_t(x_t) - \sum_{t=1}^{T} l_t(u_t) \leq \varepsilon G + |\hat{u}| \left( G + \sqrt{\sum_{t=1}^{T} \langle g_t, h_t \rangle^2} \right) \cdot \text{polylog} \left( |\hat{u}|, T, \varepsilon^{-1} \right)$$

$$= \varepsilon G + \left( G \frac{\|u_{1:T}\|_2}{\|h_{1:T}\|_2} + \sqrt{\sum_{t=1}^{T} \langle g_t, u_t \rangle^2} \right) \cdot \text{polylog} \left( \frac{\|u_{1:T}\|_2}{\|h_{1:T}\|_2}, T, \varepsilon^{-1} \right)$$

$$\leq \varepsilon G + \left( G \frac{\|u_{1:T}\|_2}{\|h_{1:T}\|_2} + \sqrt{\sum_{t=1}^{T} \langle g_t, u_t \rangle^2} \right) \cdot \text{polylog} \left( \max_t \|u_t\|_2, T, \varepsilon^{-1} \right),$$

where the last line is due to our assumption that $\|h_{1:T}\|_2 \geq 1$. $\qquad\square$

Next, we prove the unconstrained dynamic regret bound with general dictionaries (Theorem 1).

**Theorem 4** (Theorem 1, full). *Consider any collection of signals $z^{(n)} \in \text{span}(h_{1:T,n})$, $\forall n$. We define its reconstruction error (for the comparator $u_{1:T}$) as $z^{(0)} = u_{1:T} - \sum_{n=1}^{N} z^{(n)} \in \mathbb{R}^{dT}$. Then, for all $T \in \mathbb{N}_+$ and $u_{1:T} \in \mathbb{R}^{dT}$, against any adversary $\mathcal{E}$, Algorithm 2 guarantees*

$$\sum_{t=1}^{T} l_t(x_t) - \sum_{t=1}^{T} l_t(u_t) \leq \varepsilon G - \sum_{t=1}^{T} \left\langle g_t, z_t^{(0)} \right\rangle$$

$$+ \left( G \sum_{n=1}^{N} \frac{\|z^{(n)}\|_2}{\|h_{1:T,n}\|_2} + \sum_{n=1}^{N} \sqrt{\sum_{t=1}^{T} \left\langle g_t, z_t^{(n)} \right\rangle^2} \right) \cdot \text{polylog} \left( \max_{t,n} \left\| z_t^{(n)} \right\|_2, T, N, \varepsilon^{-1} \right),$$

*where $z_t^{(n)} \in \mathbb{R}^d$ is the t-th round component of the sequence $z^{(n)} \in \mathbb{R}^{dT}$.*

*Proof of Theorem 4.* The idea of this theorem is a dynamic analogue of [Cut19] to aggregate the regret bound of single direction learners. For all decomposition $u_{1:T} = \sum_{n=0}^{N} z^{(n)}$ such that $z^{(n)} \in \text{span}(h_{1:T,n})$ for all $n \in [1:T]$, we have

$$\sum_{t=1}^{T} l_t(x_t) - \sum_{t=1}^{T} l_t(u_t) \leq \langle g_{1:T}, x_{1:T} - u_{1:T} \rangle = \left\langle -g_{1:T}, z^{(0)} \right\rangle + \sum_{n=1}^{N} \left\langle g_{1:T}, w_{1:T,n} - z^{(n)} \right\rangle.$$

For the first term on the RHS, $\left\langle -g_{1:T}, z^{(0)} \right\rangle = -\sum_{t=1}^{T} \left\langle g_t, z_t^{(n)} \right\rangle$. As for the rest, we plug in Lemma C.2, with hyperparameter $\varepsilon/N$.

$$\sum_{n=1}^{N} \left\langle g_{1:T}, w_{1:T,n} - z^{(n)} \right\rangle$$

$$\leq \sum_{n=1}^{N} \left\{ \varepsilon N^{-1} G + \left( G \frac{\left\| z^{(n)} \right\|_2}{\left\| h_{1:T,n} \right\|_2} + \sqrt{\sum_{t=1}^{T} \left\langle g_t, z_t^{(n)} \right\rangle^2} \right) \cdot \mathrm{polylog}\left( \max_t \left\| z_t^{(n)} \right\|_2, T, N, \varepsilon^{-1} \right) \right\}$$

$$\leq \varepsilon G + \left( G \sum_{n=1}^{N} \frac{\left\| z^{(n)} \right\|_2}{\left\| h_{1:T,n} \right\|_2} + \sum_{n=1}^{N} \sqrt{\sum_{t=1}^{T} \left\langle g_t, z_t^{(n)} \right\rangle^2} \right) \cdot \mathrm{polylog}\left( \max_{t,n} \left\| z_t^{(n)} \right\|_2, T, N, \varepsilon^{-1} \right). \qquad \square$$

Next, we show how this dynamic regret bound recovers the static regret bound in $\mathbb{R}^d$. As discussed in Section 2.2, the static setting amounts to picking $N = d$ and $\mathcal{H}_t = I_d$, and the decomposed signals $z^{(n)}$ are determined by orthogonal projection of the static comparator sequence $u_{1:T} = [u, \ldots, u]$.

Specifically, $z_t^{(n)}$ is a $d$-dimensional vector which is zero except the $n$-th entry; its $n$-th entry equals the $n$-th entry of the static comparator $u$. If we index the gradient as $g_t = [g_{t,1}, \ldots, g_{t,d}] \in \mathbb{R}^d$ and the static comparator as $u = [u_1, \ldots, u_d] \in \mathbb{R}^d$, then $\left\langle g_t, z_t^{(n)} \right\rangle = g_{t,n} u_n$. Applying Theorem 4, against static $u_{1:T}$,

$$\sum_{t=1}^{T} l_t(x_t) - \sum_{t=1}^{T} l_t(u_t)$$

$$\leq \varepsilon G + \left( G \sum_{n=1}^{N} \frac{\left\| z^{(n)} \right\|_2}{\left\| h_{1:T,n} \right\|_2} + \sum_{n=1}^{N} \sqrt{\sum_{t=1}^{T} \left\langle g_t, z_t^{(n)} \right\rangle^2} \right) \cdot \mathrm{polylog}\left( \max_{t,n} \left\| z_t^{(n)} \right\|_2, T, N, \varepsilon^{-1} \right)$$

$$\leq \varepsilon G + \left( G \sum_{i=1}^{d} \frac{|u_i| \sqrt{T}}{\sqrt{T}} + \sum_{i=1}^{d} |u_i| \sqrt{\sum_{t=1}^{T} g_{t,i}^2} \right) \cdot \mathrm{polylog}\left( \left\| u \right\|_\infty, T, N, \varepsilon^{-1} \right)$$

$$\leq \varepsilon G + \left( G \left\| u \right\|_1 + \left\| u \right\|_2 \sqrt{\sum_{t=1}^{T} \left\| g_t \right\|_2^2} \right) \cdot \mathrm{polylog}\left( \left\| u \right\|_\infty, T, N, \varepsilon^{-1} \right). \qquad \text{(Cauchy-Schwarz)}$$

In the asymptotic regime with large $T$, $\mathrm{Regret}_T(u_{1:T}) = \tilde{O}(\left\| u \right\|_2 \sqrt{T})$.

## D  Detail on the wavelet algorithm

This section presents details of our wavelet algorithm. The pseudocode is presented in Appendix D.1. Appendix D.2 introduces the wavelet-specific notations for our analysis. Appendix D.3 presents a generic sparsity based bound for our algorithm. Appendix D.4 and D.5 prove our main results. Auxiliary lemmas are contained in Appendix D.6. Finally, Appendix D.7 works out the details of the two examples from the main paper.

### D.1  Pseudocode

For all $m \in \mathbb{N}_+$, let $T = 2^m$, and let $\mathrm{Haar}_m$ be the $T \times T$ Haar dictionary matrix defined in Section 3.1, for $d = 1$. We apply the following variant (Algorithm 4) of our sparse coding framework, in order to remove all $d$ dependence from the final regret bound. It adopts the $d$ dimensional version of the static subroutine (FREEGRAD), instead of the 1D version in Section 2. The pseudocode mirrors the combination of Algorithm 1 and 2.

It is equivalent to view Algorithm 4 as operating on a $dT \times dT$ "master" dictionary matrix $\mathcal{H}$, defined block-wise as the following: for all $(i, j) \in [1 : T]^2$, the $(i, j)$-th block of $\mathcal{H}$ is the product of the

---
**Algorithm 4** Haar OLR with known time horizon.
---
**Require:** A time horizon $T = 2^m$; the $T \times T$ Haar dictionary matrix $\text{Haar}_m$; and a hyperparameter
    $\varepsilon > 0$ (default is 1).
1: Let $N = T$. For all $n \in [1 : N]$, initialize a copy of the $d$ dimensional version of Algorithm 3
    (FREEGRAD) as $\mathcal{A}_n$, with hyperparameter $\varepsilon/N$.
2: **for** $t = 1, 2, \ldots,$ **do**
3:     Receive the $t$-th row of $\text{Haar}_m$, and index it as $[h_{t,1}, \ldots, h_{t,N}]$; note that $h_{t,n} \in \mathbb{R}$.
4:     **for** $n = 1, 2, \ldots, N$ **do**
5:         If $h_{t,n} \neq 0$, query $\mathcal{A}_n$ for its output, and assign it to $\hat{x}_{t,n} \in \mathbb{R}^d$; otherwise, $\hat{x}_{t,n}$ is arbitrary.
6:         Define $w_{t,n} = h_{t,n}\hat{x}_{t,n} \in \mathbb{R}^d$.
7:     **end for**
8:     Predict $x_t = \sum_{n=1}^{N} w_{t,n} \in \mathbb{R}^d$, receive loss gradient $g_t \in \mathbb{R}^d$.
9:     **for** $n = 1, 2, \ldots, N$ **do**
10:       If $h_{t,n} \neq 0$, compute $\hat{g}_{t,n} = h_{t,n}g_t$ and send it to $\mathcal{A}_n$ as its surrogate loss gradient.
11:     **end for**
12: **end for**
---

---
**Algorithm 5** Anytime Haar OLR (Algorithm 4 with doubling trick).
---
1: **for** $m = 1, 2, \ldots,$ **do**
2:     Run Algorithm 4 for $2^m$ rounds, which uses the matrix $\text{Haar}_m$. The hyperparameter is set to
    1.
3: **end for**
---

$(i, j)$-th entry of $\text{Haar}_m$ (which is a scalar) and the $d$-dimensional identity matrix $I_d$. That is, $\mathcal{H}$ is a block matrix; each block is a diagonal matrix with equal diagonal entries determined by $\text{Haar}_m$. Roughly, the algorithm measures distances in $\mathbb{R}^d$ by the $L_2$ norm, while measuring $\mathbb{R}^T$ by the $L_1$ norm.

Algorithm 4 alone is not sufficient for our purpose: it must take an integer $m$ and run for a fixed $T = 2^m$ rounds. We apply a meta algorithm (Algorithm 5), which simply restarts the known $T$ algorithm using the classical doubling trick, c.f., [SS11, Section 2.3.1].

### D.2 More background

Although the analysis of our framework is simpler than [JC22], a challenge is carefully indexing all the quantities to account for the vectorized setting. It is thus important to introduce a few notations to streamline the presentation. $\text{Haar}_m$ is the $T \times T$ Haar dictionary matrix defined in Section 3.1, with $T = 2^m$. Recall the statistics of the comparator sequence, summarized in Appendix A.

**Local interval** Given any scale-location pair $(j, l)$, let the support $I^{(j,l)}$ be the time interval where the feature $h^{(j,l)}$ is nonzero. That is,

$$I^{(j,l)} := [2^j(l-1) + 1 : 2^j l].$$

Moreover, let $I_+^{(j,l)}$ denote the first half of this interval, and $I_-^{(j,l)}$ for the second half. $h^{(j,l)}$ is 1 on $I_+^{(j,l)}$, and $-1$ on $I_-^{(j,l)}$.

**Normalization** Let $\widetilde{\text{Haar}}_m$ be the orthonormal matrix obtained by scaling the columns of $\text{Haar}_m$. The normalized feature vectors are also denoted by tilde, i.e., instead of $h^*$ and $h^{(j,l)}$, the normalized features are $\tilde{h}^*$ and $\tilde{h}^{(j,l)}$. They are vectors in $\mathbb{R}^T$, with the $t$-th component denoted by $\tilde{h}_t^*$ and $\tilde{h}_t^{(j,l)}$, in $\mathbb{R}$.

**Coordinate sequence** Consider any comparator sequence $u_{1:T} \in \mathbb{R}^{dT}$. For all coordinate $i \in [1 : d]$, we define its $i$-th coordinate sequence as $u_{1:T}^{(i)} \in \mathbb{R}^T$: the $t$-th entry of this coordinate sequence $u_{1:T}^{(i)}$, denoted by $u_t^{(i)}$, is the $i$-th coordinate of $u_t$.

**Transform domain coefficient** We will also use the transform domain coefficients of $u_{1:T}$, under the Haar wavelet transform. Recall that in the single-feature, generic setting (Section 2.2), we denoted a single transform domain coefficient by $\hat{u} \in \mathbb{R}$. With wavelets, the transform domain encodes $dT$-dimensional vectors. According to our convention so far, we will denote them by scale-location pairs $(j, l)$: given a $(j, l)$ pair, the "coefficient" $\hat{u}^{(j,l)}$ is a $d$-dimensional vector. There are $T - 1$ pairs of $(j, l)$ in total; complementing the representation, we use another $\hat{u}^* \in \mathbb{R}^d$ to represent the "coefficient" for the all-one feature.

Given any scale parameter $j \in [1 : \log_2 T]$ and location parameter $l \in [1 : 2^{-j}T]$, let

$$\hat{u}^{(j,l)} := \left[ \left\langle \tilde{h}^{(j,l)}, u_{1:T}^{(1)} \right\rangle, \dots, \left\langle \tilde{h}^{(j,l)}, u_{1:T}^{(d)} \right\rangle \right],$$

and for the all-one feature,

$$\hat{u}^* := \left[ \left\langle \tilde{h}^*, u_{1:T}^{(1)} \right\rangle, \dots, \left\langle \tilde{h}^*, u_{1:T}^{(d)} \right\rangle \right].$$

That is, each entry is the inner product between the normalized feature and a coordinate sequence from $u_{1:T}$.

Due to the orthonormality of the applied transform (specified by the normalized features $\tilde{h}^*$ and $\tilde{h}^{(j,l)}$), the energy is preserved between the time domain and the transform domain, i.e.,

$$E = \|u_{1:T}\|_2^2 = \|\hat{u}^*\|_2^2 + \sum_{j,l} \left\| \hat{u}^{(j,l)} \right\|_2^2,$$

and also the second order variability (the energy of the centered dynamic component within $u_{1:T}$),

$$\bar{E} = \sum_{t=1}^{T} \|u_t - \bar{u}\|_2^2 = \sum_{j,l} \left\| \hat{u}^{(j,l)} \right\|_2^2. \tag{11}$$

Moreover, since $\tilde{h}^*$ equals $1/\sqrt{T}$ times the all-one vector,

$$\|\hat{u}^*\|_2^2 = \sum_{i=1}^{d} \left\langle \tilde{h}^*, u_{1:T}^{(i)} \right\rangle^2 = \sum_{i=1}^{d} \left( \frac{1}{\sqrt{T}} \sum_t u_{1:T}^{(i)} \right)^2 = T \sum_{i=1}^{d} \left( \frac{1}{T} \sum_t u_{1:T}^{(i)} \right)^2 = \|\bar{u}\|_2^2 \, T. \tag{12}$$

**Detail reconstruction** Given the transform domain coefficients, we can reconstruct details of the comparator $u_{1:T}$ on the time domain. Similar to our notation in the generic framework (Section 2.2), we keep the letter $z$, but replace the index $n$ by $(j, l)$, which is more suitable for indexing wavelets.

Let $z^{(j,l)} \in \mathbb{R}^{dT}$ be the detail of $u_{1:T}$ along the $(j, l)$-th feature. It is the concatenation of $T$ vectors in $\mathbb{R}^d$, and for all $t$, the $t$-th of these vectors is defined by

$$z_t^{(j,l)} := \hat{u}^{(j,l)} \tilde{h}_t^{(j,l)} \in \mathbb{R}^d.$$

Similarly, we can define the detail $z^*$ along the feature $\tilde{h}^*$. Its $t$-th component is

$$z_t^* := \hat{u}^* \tilde{h}_t^*,$$

and clearly, the RHS does not depend on $t$ since $\tilde{h}^*$ is the normalization of the all-one feature $h^*$.

Let us also sum the details across different locations. Given a scale $j$, let

$$z^{(j)} := \sum_l z^{(j,l)} \in \mathbb{R}^{dT}.$$

Note that the summands are sequences that do not overlap: at each entry, only one of the summand sequence is nonzero. The full reconstruction is obtained by summing all the details,

$$u_{1:T} := z^* + \sum_{j=1}^{\log_2 T} z^{(j)}.$$

**Statistics of the detail sequence**  We can define statistics of the detail sequences just like the statistics of the comparator $u_{1:T}$. Specifically, define the first order variability of the $(j, l)$-th detail as

$$\bar{S}^{(j,l)} := \sum_{t=1}^{T} \left\| z_t^{(j,l)} \right\|_2.$$

Note that since the $z_t^{(j,l)}$ sequence is centered (with average being equal to 0), its first order variability equals its norm sum, c.f., Appendix A. Summing over the locations, the first order variability at the $j$-th scale is

$$\bar{S}^{(j)} := \sum_{t=1}^{T} \left\| z_t^{(j)} \right\|_2,$$

which equals $\sum_l \bar{S}^{(j,l)}$.

Similarly, we can define the path length of the detail sequences. A caveat is that we only count the path length *within* the support $I^{(j,l)}$ of the feature $h^{(j,l)}$,

$$P^{(j,l)} := \sum_{t=2^j(l-1)+1}^{2^j l - 1} \left\| z_{t+1}^{(j,l)} - z_t^{(j,l)} \right\|_2.$$

The comparator's movement when the support changes does not count. Summing over the locations,

$$P^{(j)} := \sum_l P^{(j,l)}.$$

## D.3  Generic sparsity adaptive bound

With the notation from the previous subsection, we now present a generic sparsity adaptive regret bound for Algorithm 4 (fixed $T$ Haar OLR). Since the latter is a variant of our main sparse coding framework (Section 2), the result can be analogously derived, although the notations need to be treated carefully.

**Lemma D.1.** *For any $m$, $T = 2^m$ and $u_{1:T} \in \mathbb{R}^{dT}$, with any hyperparameter $\varepsilon > 0$, Algorithm 4 guarantees*

$$\mathrm{Regret}_T(u_{1:T}) \le \varepsilon G + G \left( \|z^*\|_2 + \sum_{j=1}^{\log_2 T} \sum_{l=1}^{2^{-j}T} \left\| z^{(j,l)} \right\|_2 \right) \cdot \mathrm{polylog}\left(M, T, \varepsilon^{-1}\right).$$

The proof sums the regret bound of the $d$-dimensional version of the static subroutine (Lemma C.1), across $T$ different copies. It is very similar to Theorem 1, therefore omitted.

It might be more convenient to use the transform domain coefficients $\hat{u}^{(j,l)}$ in the bound, rather than the reconstructed details $z^{(j,l)}$. In this case, we have

$$\left\| z^{(j,l)} \right\|_2^2 = \sum_t \left\| z_t^{(j,l)} \right\|_2^2 = \sum_t \left[ \left\| \hat{u}^{(j,l)} \right\|_2^2 \left| \tilde{h}_t^{(j,l)} \right|^2 \right] = \left\| \hat{u}^{(j,l)} \right\|_2^2 \sum_t \left| \tilde{h}_t^{(j,l)} \right|^2 = \left\| \hat{u}^{(j,l)} \right\|_2^2.$$

Similarly,

$$\|z^*\|_2^2 = \|\hat{u}^*\|_2^2.$$

Therefore,

$$\mathrm{Regret}_T(u_{1:T}) \le \varepsilon G + G \left( \|\hat{u}^*\|_2 + \sum_{j=1}^{\log_2 T} \sum_{l=1}^{2^{-j}T} \left\| \hat{u}^{(j,l)} \right\|_2 \right) \cdot \mathrm{polylog}\left(M, T, \varepsilon^{-1}\right). \tag{13}$$

## D.4  Unconstrained switching regret

In the $K$-switching regret, the complexity of the comparator is characterized by its amount of switches. The idea is that, if the comparator $u_{1:T}$ is static on a support $I^{(j,l)}$ for some $(j, l)$, then the corresponding transform domain coefficient $\hat{u}^{(j,l)} = 0 \in \mathbb{R}^d$. We have the following bound for the fixed $T$ algorithm (Algorithm 4).

**Lemma D.2.** *For any $m$, $T = 2^m$ and $u_{1:T} \in \mathbb{R}^{dT}$, Algorithm 4 with the hyperparameter $\varepsilon = 1$ guarantees*

$$\text{Regret}_T(u_{1:T}) = \tilde{O}\left(\|\bar{u}\|_2 \sqrt{T} + \sqrt{K\bar{E}}\right).$$

*Proof of Lemma D.2.* Consider any scale $j$. Since the supports $\{I^{(j,l)}\}_l$ do not overlap, if $u_{1:T}$ shifts $K$ times, then there are at most $K$ choices of location $l$ such that the transform domain coefficient $\hat{u}^{(j,l)}$ is nonzero. Furthermore, since there are $\log_2 T$ scales in total, there are at most $K \log_2 T$ pairs of $(i, l)$ such that $\hat{u}^{(j,l)}$ is nonzero. Therefore, using Cauchy-Schwarz and Eq.(11),

$$\sum_{j,l} \left\|\hat{u}^{(j,l)}\right\|_2 \leq \sqrt{K\log_2 T}\sqrt{\sum_{j,l}\left\|\hat{u}^{(j,l)}\right\|_2^2} = \sqrt{K\bar{E}\log_2 T}.$$

Plugging this into Eq.(13), and further using Eq.(12) for $\|\hat{u}^*\|_2$ complete the proof. $\qquad\square$

The anytime bound in general follows from the classical doubling trick. A twist is that the analysis is slightly more involved than the standard one, e.g., [SS11, Section 2.3.1], as we also need to relate the comparator statistics on each block to those for the entire signal $u_{1:T}$.

**Theorem 2** (Switching regret). *For all $T \in \mathbb{N}_+$ and $u_{1:T} \in \mathbb{R}^{dT}$, Algorithm 5 guarantees*

$$\text{Regret}_T(u_{1:T}) \leq \tilde{O}\left(\|\bar{u}\|_2 \sqrt{T} + \sqrt{K\bar{E}}\right). \tag{9}$$

*Proof of Theorem 2.* First, assume $T$ can be exactly decomposed into $m^*$ segments with dyadic lengths $2^1, \ldots, 2^{m^*}$. We use $\bar{u}_m$, $K_m$ and $\bar{E}_m$ to represent the statistics of the comparator sequence on the length $2^m$ block, and let $I^m$ denote the time interval that this block operates on. $\bar{u}$, $K$ and $S$ denote the statistics of the entire signal $u_{1:T}$, c.f., Appendix A. From Lemma D.2,

$$\text{Regret}_T(u_{1:T}) \leq \sum_{m=1}^{m^*} \tilde{O}\left(\|\bar{u}_m\|_2\sqrt{2^m} + \sqrt{K_m\bar{E}_m}\right)$$

$$\leq \tilde{O}\left[\|\bar{u}\|_2\left(\sum_{m=1}^{m^*}\sqrt{2^m}\right) + \sum_{m=1}^{m^*}\|\bar{u}_m - \bar{u}\|_2\sqrt{2^m} + \sum_{m=1}^{m^*}\sqrt{K_m\bar{E}_m}\right]. \tag{14}$$

The first term follows from the standard doubling trick analysis [SS11, Section 2.3.1],

$$\sum_{m=1}^{m^*}\sqrt{2^m} \leq \frac{\sqrt{2}}{\sqrt{2}-1}\sqrt{2^{m^*}} = O\left(\sqrt{T}\right). \tag{15}$$

As for the second term in Eq.(14), using Cauchy-Schwarz,

$$\sum_{m=1}^{m^*}\|\bar{u}_m - \bar{u}\|_2\sqrt{2^m} \leq \sqrt{m^*\left(\sum_{m=1}^{m^*} 2^m\|\bar{u}_m - \bar{u}\|_2^2\right)}.$$

$m^* = O(\log T)$, and also observe that the sum (in the parenthesis) on the RHS equals the second order variability of the following signal: for any time $t$ in the $m$-th block, the signal's component is $\bar{u}_m \in \mathbb{R}^d$. This signal is a locally averaged version of the original comparator $u_{1:T}$, and the key idea is that local averaging decreases the variability. Formally, due to Lemma D.7, we have

$$\sum_{m=1}^{m^*}\|\bar{u}_m - \bar{u}\|_2\sqrt{2^m} \leq \tilde{O}\left(\sqrt{\bar{E}}\right). \tag{16}$$

For the third term in Eq.(14), using Cauchy-Schwarz again,

$$\sum_{m=1}^{m^*}\sqrt{K_m\bar{E}_m} \leq \sqrt{\left(\sum_{m=1}^{m^*}K_m\right)\left(\sum_{m=1}^{m^*}\bar{E}_m\right)} \leq \sqrt{K\bar{E}}.$$

The sum of $K_m$ is straightforward. The inequality for the sum of $\bar{E}_m$ follows from the observation that on the $m$-th block, $\bar{u}_m$ minimizes $\sum_{t \in I^m} \|u_t - x\|_2^2$ with respect to $x \in \mathbb{R}^d$.

Also, notice that the second term in Eq.(14) is dominated by the third term. If $K = 0$, then both $\sqrt{\bar{E}}$ and $\sqrt{K\bar{E}}$ equal 0. If $K \geq 1$, then $\sqrt{\bar{E}} \leq \sqrt{K\bar{E}}$. Therefore, Eq.(14) can be written as

$$\text{Regret}_T(u_{1:T}) \leq \tilde{O}\left(\|\bar{u}\|_2 \sqrt{T} + \sqrt{K\bar{E}}\right).$$

As for the general setting where $T$ cannot be exactly decomposed into dyadic blocks: consider the smallest $T^* > T$ such that $T^*$ can be decomposed. Due to doubling intervals, $T^* \leq 2T$. Let us consider a hypothetical length $T^*$ game with the rounds $t > T$ constructed as follows: the loss gradient $g_t = 0 \in \mathbb{R}^d$, and $u_t = \bar{u}$. In this case, with $K$ and $\bar{E}$ still representing the statistics of the length $T$ sequence $u_{1:T}$, the number of switches on the entire time interval $[1 : T^*]$ is at most $K + 1$, and the second order variability on $[1 : T^*]$ is $\bar{E}$; furthermore, it is clear that $(K + 1)\bar{E} \leq 2K\bar{E}$. The regret of any algorithm on this hypothetical length $T^*$ game is the same as the length $T$ game, therefore bounding the latter follows from bounding the former. $\qquad\square$

### D.5  Path-length-based bound

Next, we turn to bounds that depend on the path length $P$ of the comparator $u_{1:T}$. Similar to the switching regret analysis, we will first consider the setting with fixed dyadic $T$ (Algorithm 4), and then extend its guarantee through a doubling trick.

#### D.5.1  Fixed dyadic horizon

In the following, we consider Algorithm 4; assume $T = 2^m$ for some $m$. The static component (i.e., $z^*$) and the dynamic component (i.e., $u_{1:T} - z^*$) of $u_{1:T}$ are analyzed separately; the former is fairly standard, while the latter is more challenging. We will first consider the dynamic component, and proceed in three steps.

**Step 1**  Considering any scale $j$, we aim to show $\sum_l \left\|\hat{u}^{(j,l)}\right\|_2 \leq \sqrt{P^{(j)}\bar{S}^{(j)}}$, which relates the transform domain coefficients to the regularity of the reconstructed signals.

**Lemma D.3.** *For all $(j, l)$ pair,*

$$\left\|\hat{u}^{(j,l)}\right\|_2 = 2^{-1/2}\sqrt{P^{(j,l)}\bar{S}^{(j,l)}},$$

*and*

$$\sum_l \left\|\hat{u}^{(j,l)}\right\|_2 \leq 2^{-1/2}\sqrt{P^{(j)}\bar{S}^{(j)}}.$$

*Proof of Lemma D.3.*  Let us start from the first part of this lemma, and express the detail sequence $z^{(j,l)}$, and equivalently $z^{(j)}$, more explicitly on its support $I^{(j,l)}$.

$$z_t^{(j)} = \begin{cases} 2^{-j/2}\hat{u}^{(j,l)}, & t \in I_+^{(j,l)}; \\ -2^{-j/2}\hat{u}^{(j,l)}, & t \in I_-^{(j,l)}. \end{cases}$$

Rewriting $P^{(j,l)}$ and $\bar{S}^{(j,l)}$,

$$P^{(j,l)} = \sum_{t=2^j(l-1)+1}^{2^j l - 1} \left\|z_{t+1}^{(j)} - z_t^{(j)}\right\|_2 = 2^{1-j/2}\left\|\hat{u}^{(j,l)}\right\|_2.$$

$$\bar{S}^{(j,l)} = \sum_{t \in I^{(j,l)}} \left\|z_t^{(j)}\right\|_2 = 2^{-j/2}\left\|\hat{u}^{(j,l)}\right\|_2 \cdot 2^j = 2^{j/2}\left\|\hat{u}^{(j,l)}\right\|_2,$$

which yields the equality in the lemma. The second part follows from Cauchy-Schwarz. $\qquad\square$

**Step 2** Showing that $P^{(j)} \leq P$ and $\bar{S}^{(j)} \leq \bar{S}$. That is, the reconstructed signals are easier than the original comparator $u_{1:T}$. Here, $P$ and $\bar{S}$ should be considered separately.

**Lemma D.4.** *For any $u_{1:T}$ and any scale parameter $j^*$, $P^{(j^*)} \leq P$.*

*Proof of Lemma D.4.* From the definition of $P$ and the reconstruction of $u_{1:T}$ from detail sequences,

$$P = \sum_{t=1}^{T-1} \|u_{t+1} - u_t\|_2 = \sum_{t=1}^{T-1} \left\| z_{t+1}^* - z_t^* + \sum_j \left( z_{t+1}^{(j)} - z_t^{(j)} \right) \right\|_2 = \sum_{t=1}^{T-1} \left\| \sum_j \left( z_{t+1}^{(j)} - z_t^{(j)} \right) \right\|_2,$$

where the last equality is due to $z^*$ being a constant sequence.

Consider removing "shorter" scales with $1 \leq j < j^*$, which is equivalent to local averaging, c.f., Appendix D.6. Due to Lemma D.7, the path length does not increase, i.e,

$$\sum_{t=1}^{T-1} \left\| \sum_j \left( z_{t+1}^{(j)} - z_t^{(j)} \right) \right\|_2 \geq \sum_{t=1}^{T-1} \left\| \sum_{j \geq j^*} \left( z_{t+1}^{(j)} - z_t^{(j)} \right) \right\|_2.$$

Then, we can further remove the rounds where the path length is not counted in $P^{(j^*)}$, i.e., when a time $t \in I^{(j^*,l)}$ but $t+1 \in I^{(j^*,l+1)}$.

$$\text{RHS} \geq \sum_l \sum_{t=2^{j^*}(l-1)+1}^{2^{j^*}l-1} \left\| \sum_{j \geq j^*} \left( z_{t+1}^{(j)} - z_t^{(j)} \right) \right\|_2.$$

Now, consider any location $l$, which determines the time interval $I^{(j^*,l)} = [2^{j^*}(l-1)+1 : 2^{j^*}l]$. Any detail sequence $z^{(j)}$ with scale $j > j^*$ is constant on this time interval, thus removing it does not change the path length at all. Therefore,

$$P \geq \sum_l \sum_{t=2^{j^*}(l-1)+1}^{2^{j^*}l-1} \left\| z_{t+1}^{(j^*)} - z_t^{(j^*)} \right\|_2 = P^{(j^*)}. \qquad \square$$

As for the first order variability,

**Lemma D.5.** *For any $u_{1:T}$ and any scale parameter $j^*$, $\bar{S}^{(j^*)} \leq \bar{S}$.*

*Proof of Lemma D.5.* From the definition, noticing that $\bar{u}$ is entirely captured by the all-one feature,

$$\bar{S} = \sum_{t=1}^{T} \|u_t - \bar{u}\|_2 = \sum_{t=1}^{T} \left\| \sum_{j=1}^{\log_2 T} z_t^{(j)} \right\|_2.$$

Due to Lemma D.7, removing short scales amounts to local averaging, which decreases the variability.

$$\bar{S} \geq \sum_{t=1}^{T} \left\| \sum_{j \geq j^*} z_t^{(j)} \right\|_2 = \sum_l \sum_{t \in I^{(j^*,l)}} \left\| z_t^{(j^*)} + \sum_{j > j^*} z_t^{(j)} \right\|_2.$$

For any $l$, consider the support of the $(j^*, l)$-th feature, $I^{(j^*,l)}$. Observe that $\sum_{j > j^*} z_t^{(j)}$ is time invariant throughout $I^{(j^*,l)}$, let us denote it as $v \in \mathbb{R}^d$. Meanwhile, for some $w \in \mathbb{R}^d$, $z_t^{(j^*)}$ equals $w$ on $I_+^{(j^*,l)}$, the first half of this interval, while being $-w$ on the second half $I_-^{(j^*,l)}$ of this interval. Therefore,

$$\sum_{t \in I^{(j^*,l)}} \left\| z_t^{(j^*)} + \sum_{j > j^*} z_t^{(j)} \right\|_2 = 2^{j^*-1} \left( \|v + w\|_2 + \|v - w\|_2 \right) \geq 2^{j^*} \|w\|_2 = \sum_{t \in I^{(j^*,l)}} \left\| z_t^{(j^*)} \right\|_2.$$

Combining the above,

$$\bar{S} \geq \sum_l \sum_{t \in I^{(j^*,l)}} \left\| z_t^{(j^*)} \right\|_2 = \bar{S}^{(j^*)}. \qquad \square$$

**Step 3** Summarizing the above relations, and using the property that there are only $\log_2 T$ scales.

**Lemma D.6.** *For any $m$, $T = 2^m$ and $u_{1:T} \in \mathbb{R}^{dT}$, Algorithm 4 with the hyperparameter $\varepsilon = 1$ guarantees*

$$\text{Regret}_T(u_{1:T}) = \tilde{O}\left(\|\bar{u}\|_2 \sqrt{T} + \sqrt{P\bar{S}}\right).$$

*Proof of Lemma D.6.* Starting from the generic regret bound, Eq.(13) for Algorithm 4.

$$\text{Regret}_T(u_{1:T}) \leq \varepsilon G + G\left(\|\hat{u}^*\|_2 + \sum_{j,l}\left\|\hat{u}^{(j,l)}\right\|_2\right) \cdot \text{polylog}\left(M, T, \varepsilon^{-1}\right).$$

Due to Eq.(12), $\|\hat{u}^*\|_2 = \|\bar{u}\|_2 \sqrt{T}$. Then, combining Lemma D.3, D.4 and D.5,

$$\sum_{j,l}\left\|\hat{u}^{(j,l)}\right\|_2 \leq O\left(\sqrt{P\bar{S}}\log_2 T\right).$$

Plugging it into the generic bound completes the proof. $\square$

### D.5.2 Anytime bound

Now we are ready to prove an anytime unconstrained dynamic regret bound that depends on the path length.

**Theorem 3** (Path length bound). *For all $T \in \mathbb{N}_+$ and $u_{1:T} \in \mathbb{R}^{dT}$, Algorithm 5 guarantees*

$$\text{Regret}_T(u_{1:T}) \leq \tilde{O}\left(\|\bar{u}\|_2 \sqrt{T} + \sqrt{P\bar{S}}\right). \tag{10}$$

*Proof of Theorem 3.* Similar to the analysis of the switching regret (Theorem 2), we first consider the situation where the time horizon $T$ can be exactly decomposed into $m^*$ segments with dyadic lengths $2^1, \ldots, 2^{m^*}$. In this situation, we have

$$\text{Regret}_T(u_{1:T}) \leq \tilde{O}\left[\sum_{m=1}^{m^*} \|\bar{u}_m\|_2 \sqrt{2^m} + \sum_{m=1}^{m^*} \sqrt{P_m \bar{S}_m}\right]$$

$$\leq \tilde{O}\left[\|\bar{u}\|_2 \sqrt{T} + \sqrt{\bar{E}} + \sum_{m=1}^{m^*} \sqrt{P_m \bar{S}_m}\right],$$

where the second line follows from the proof of Theorem 2, specifically Eq.(15) and Eq.(16).

Now let us consider the remaining sum on the RHS. Using Cauchy-Schwarz,

$$\sum_{m=1}^{m^*} \sqrt{P_m \bar{S}_m} \leq \sqrt{\left(\sum_{m=1}^{m^*} P_m\right)\left(\sum_{m=1}^{m^*} \bar{S}_m\right)} \leq \sqrt{P\left(\sum_{m=1}^{m^*} \sum_{t=2^m-1}^{2^{m+1}-2} \|u_t - \bar{u}_m\|_2\right)},$$

where

$$\sum_{m=1}^{m^*} \sum_{t=2^m-1}^{2^{m+1}-2} \|u_t - \bar{u}_m\|_2 \leq \sum_{m=1}^{m^*} \sum_{t=2^m-1}^{2^{m+1}-2} \left(\|u_t - \bar{u}\|_2 + \|\bar{u}_m - \bar{u}\|_2\right) = \bar{S} + \sum_{m=1}^{m^*} 2^m \|\bar{u}_m - \bar{u}\|_2.$$

The last sum on the RHS is the first order variability of a locally averaged version of $u_{1:T}$. Due to Lemma D.7,

$$\sum_{m=1}^{m^*} 2^m \|\bar{u}_m - \bar{u}\|_2 \leq \bar{S}.$$

Combining everything above,

$$\text{Regret}_T(u_{1:T}) \leq \tilde{O}\left(\|\bar{u}\|_2 \sqrt{T} + \sqrt{\bar{E}} + \sqrt{P\bar{S}}\right).$$

It remains to show that $\sqrt{\bar{E}} \le \sqrt{P\bar{S}}$, thus the former can be absorbed into the latter. Plugging in the definitions, this is equivalent to showing

$$\sum_{t=1}^{T} \|u_t - \bar{u}\|_2^2 \le \sum_{t=1}^{T} P \|u_t - \bar{u}\|_2,$$

and it suffices to prove $\|u_t - \bar{u}\| \le P$ for all $t \in [1:T]$. This is completed in Lemma D.8. Till this point, we have shown the desirable result in the situation of "exact dyadic partitioning".

To complete the proof, we turn to the general situation where $T$ cannot be partitioned into dyadic blocks. This follows from a similar "padding" construction from the proof of Theorem 2. Let $T^* = 2^{\lceil \log_2 T \rceil}$, and by definition, $T^* \le 2T$. Let us consider a hypothetical length $T^*$ game with the rounds $t > T$ constructed as follows: the loss gradient $g_t = 0 \in \mathbb{R}^d$, and $u_t = \bar{u}$. Then, the regret of any algorithm on the length $T^*$ hypothetical game equals its regret on the actual length $T$ game, and the regret bound for the former applies to the latter as well: if we write $P^*$ and $\bar{S}^*$ as the statistics of the extended length $T^*$ comparator, then

$$\text{Regret}_T(u_{1:T}) \le \tilde{O}\left(\|\bar{u}\|_2 \sqrt{T^*} + \sqrt{P^*\bar{S}^*}\right).$$

Clearly, $\bar{S}^* = \bar{S}$ and $T^* \le 2T$. As for the path length, $P^* = P + \|u_T - \bar{u}\|_2$, and due to Lemma D.8, $\|u_T - \bar{u}\|_2 \le P$. Plugging it back completes the proof. $\qquad\square$

### D.6 Useful lemma

Our analysis uses two auxiliary lemmas. First, we show that local averaging makes a signal "more regular". Consider any signal $u_{1:T} \in \mathbb{R}^{dT}$, with the $t$-th round component $u_t \in \mathbb{R}^d$. Local averaging refers to replacing any $k$ consecutive components of $u_{1:T}$ by their average, i.e., setting

$$u_{\tau+1}, \ldots, u_{\tau+k} = k^{-1} \sum_{i=1}^{k} u_{\tau+i},$$

for some $\tau \in [0:T-k]$.

**Lemma D.7.** *Let a signal $w_{1:T} \in \mathbb{R}^{dT}$ be the result of $u_{1:T}$ after local averaging, and $\bar{w} = T^{-1} \sum_{t=1}^{T} w_t \in \mathbb{R}^d$. Then, the path length, the norm sum and the energy of $w_{1:T}$, including their centered versions, are all dominated by those of $u_{1:T}$. That is,*

*1. $\sum_{t=1}^{T-1} \|w_{t+1} - w_t\|_2 \le \sum_{t=1}^{T-1} \|u_{t+1} - u_t\|_2$;*

*2. $\sum_{t=1}^{T} \|w_t - \bar{w}\|_2 \le \sum_{t=1}^{T} \|u_t - \bar{u}\|_2$;*

*3. $\sum_{t=1}^{T} \|w_t - \bar{w}\|_2^2 \le \sum_{t=1}^{T} \|u_t - \bar{u}\|_2^2$.*

*4. $\sum_{t=1}^{T} \|w_t\|_2 \le \sum_{t=1}^{T} \|u_t\|_2$, and $\sum_{t=1}^{T} \|w_t\|_2^2 \le \sum_{t=1}^{T} \|u_t\|_2^2$.*

*Proof of Lemma D.7.* Starting from the first part of the lemma, we prove for the general case of $0 < \tau < T - k$. The boundary cases ($\tau = 0$ and $\tau = T - k$) are analogous.

Local averaging only affects the path length caused by the averaged entries $u_{\tau+1}, \ldots, u_{\tau+k}$, and the two entries $u_\tau$ and $u_{\tau+k+1}$ right besides averaging boundary; this original path length quantity in

$u_{1:T}$ is $\sum_{i=0}^{k} \|u_{\tau+i+1} - u_{\tau+i}\|_2$. After averaging, the path length among these entries becomes

$$\left\| u_\tau - k^{-1} \sum_{i=1}^{k} u_{\tau+i} \right\|_2 + \left\| k^{-1} \sum_{i=1}^{k} u_{\tau+i} - u_{\tau+k+1} \right\|_2$$

$$= k^{-1} \left\| \sum_{i=1}^{k} (u_\tau - u_{\tau+i}) \right\|_2 + k^{-1} \left\| \sum_{i=1}^{k} (u_{\tau+i} - u_{\tau+k+1}) \right\|_2$$

$$\leq k^{-1} \sum_{i=1}^{k} \left( \|u_\tau - u_{\tau+i}\|_2 + \|u_{\tau+i} - u_{\tau+k+1}\|_2 \right)$$

$$\leq k^{-1} \sum_{i=1}^{k} \left( \sum_{j=0}^{k} \|u_{\tau+j+1} - u_{\tau+j}\|_2 \right)$$

$$= \sum_{i=0}^{k} \|u_{\tau+i+1} - u_{\tau+i}\|_2 .$$

Now consider the second part of the lemma. After local averaging, $\bar{w} = \bar{u}$. The affected part of the signal contributes to the following first order variability

$$\sum_{t=1}^{k} \|w_{\tau+i} - \bar{w}\|_2 = k \left\| k^{-1} \sum_{i=1}^{k} u_{\tau+i} - \bar{u} \right\|_2 = \left\| \sum_{i=1}^{k} u_{\tau+i} - k\bar{u} \right\|_2 \leq \sum_{t=1}^{k} \|u_{\tau+i} - \bar{u}\|_2 .$$

As for the third part of the lemma,

$$\sum_{t=1}^{k} \|w_{\tau+i} - \bar{w}\|_2^2 = k \left\| k^{-1} \sum_{i=1}^{k} u_{\tau+i} - \bar{u} \right\|_2^2 \leq k^{-1} \left( \sum_{i=1}^{k} \|u_{\tau+i} - \bar{u}\|_2 \right)^2 \leq \sum_{t=1}^{k} \|u_{\tau+i} - \bar{u}\|_2^2 ,$$

where the last inequality is due to AM-QM inequality.

The final part of the proof is the uncentered version of Part 2 and 3, which follows the same steps. In fact, any fixed reference point (for the variability) works, i.e., for all $v \in \mathbb{R}^d$,

$$\sum_{t=1}^{T} \|w_t - v\|_2 \leq \sum_{t=1}^{T} \|u_t - v\|_2 ,$$

$$\sum_{t=1}^{T} \|w_t - v\|_2^2 \leq \sum_{t=1}^{T} \|u_t - v\|_2^2 . \qquad \square$$

We also use another simple lemma.

**Lemma D.8.** *Consider any comparator sequence $u_{1:T}$. For all t, we have $\|u_t - \bar{u}\|_2 \leq P$.*

*Proof of Lemma D.8.* Starting from the definition,

$$\|u_t - \bar{u}\|_2 = \left\| u_t - \sum_{i=1}^{T} T^{-1} u_i \right\|_2 \leq T^{-1} \sum_{i=1}^{T} \|u_t - u_i\|_2 ,$$

and for all $i, t \in [1:T]$, $\|u_t - u_i\|_2 \leq P$ due to triangle inequality. $\qquad \square$

### D.7 Quantitative example

This subsection presents details of our two quantitative examples, Example 1 and 2.

**Tracking outliers**  We first calculate the relevant statistics of the comparator $u_{1:T}$. Note that we assume $k \le \sqrt{T}$, and in this way, there is only a small amount of $u_t$ with large magnitude, which can then be called outliers.

$$\bar{u} = \frac{1}{T}\left(k\sqrt{T} + T - k\right),$$

$|\bar{u}| = \Theta(1)$, $M = \sqrt{T}$, $K = 1$ or $2$, $P = \Theta(\sqrt{T})$, $\bar{E} \le E = kT + T - k = \Theta(kT)$. As for $S$ and $\bar{S}$,

$$S = k\sqrt{T} + T - k = \Theta(T),$$
$$\bar{S} = k(\sqrt{T} - \bar{u}) + (T - k)(\bar{u} - 1)$$
$$= 2kT^{-1}(T - k)(\sqrt{T} - 1)$$
$$\le O(k\sqrt{T}).$$

Intuitively, we have $|\bar{u}| = \Theta(1)$ while $M = \sqrt{T}$; $\bar{S} = O(k\sqrt{T})$ while $S = \Theta(T)$. This explains the improvements detailed next. For each algorithm considered in Table 1, we evaluate both its switching regret bound and its path-length-dependent bound.

- The minimax algorithm ADER [ZLZ18] is not applicable, as $M$ grows with $T$ and can be larger than any fixed diameter $D$.
- The $P$-dependent bound of the coarse baseline [JC22, Algorithm 6], c.f., Eq.(3), is
$$\tilde{O}\left(\sqrt{(M + P)MT}\right) = \tilde{O}(T).$$
  With $P = O(KM)$, the resulting $K$-dependent bound is
$$\tilde{O}\left(M\sqrt{(1 + K)T}\right) = \tilde{O}(T).$$
- The $P$-dependent bound of the fine baseline [JC22, Algorithm 2], c.f., Eq.(4), is
$$\tilde{O}\left(\sqrt{(M + P)S}\right) = \tilde{O}\left(T^{3/4}\right).$$
  With $P = O(KM)$, the resulting $K$-dependent bound is
$$\tilde{O}\left(\sqrt{(1 + K)MS}\right) = \tilde{O}(T^{3/4}).$$
- Our path length bound is
$$\tilde{O}\left(|\bar{u}|\sqrt{T} + \sqrt{P\bar{S}}\right) = \tilde{O}\left(\sqrt{kT}\right).$$
  Same for our switching regret bound,
$$\tilde{O}\left(|\bar{u}|\sqrt{T} + \sqrt{K\bar{E}}\right) = \tilde{O}\left(\sqrt{kT}\right).$$

**Persistent oscillation**  Again, we calculate the statistics of the comparator $u_{1:T}$. $\bar{u} = 1$, $M \le 2$, $K = k$, $P = \Theta(k/\sqrt{T})$. Crucially, $\bar{S} = \sqrt{T}$ and $\bar{E} = \Theta(1)$, while $S = \Theta(T)$ and $E = \Theta(T)$. Here the $K$-dependent bounds of the baselines are loose compared to their corresponding $P$-dependent bounds, due to using the relation $P = O(KM)$. Therefore we will only evaluate their $P$-dependent bounds.

- Suppose one knows that $M \le 2$ beforehand, then ADER can be applied with $D = 2$. The regret bound is
$$\tilde{O}\left(\sqrt{(D + P)DT}\right) = \tilde{O}\left(\sqrt{T} + k^{1/2}T^{1/4}\right).$$
- One could check that the $P$-dependent bounds of the coarse and the fine baselines are also
$$\tilde{O}\left(\sqrt{(M + P)MT}\right) = \tilde{O}\left(\sqrt{T} + k^{1/2}T^{1/4}\right),$$
$$\tilde{O}\left(\sqrt{(M + P)S}\right) = \tilde{O}\left(\sqrt{T} + k^{1/2}T^{1/4}\right).$$
- For our algorithm, the $P$-dependent bound is
$$\tilde{O}\left(|\bar{u}|\sqrt{T} + \sqrt{P\bar{S}}\right) = \tilde{O}\left(\sqrt{T}\right).$$
  The $K$-dependent bound is
$$\tilde{O}\left(|\bar{u}|\sqrt{T} + \sqrt{K\bar{E}}\right) = \tilde{O}\left(\sqrt{T}\right).$$

# E  Application: Time series forecasting

This section presents an application of our framework in time series forecasting.[15] Roughly speaking, we aim to address the following question:

  Given a *black box* forecaster, can we make it provably robust against (structured) nonstationarity?

Along the way, our objective is to show that

- Simultaneously handling unconstrained domains and dynamic comparators in online learning brings downstream benefits in time series forecasting.
- Our sparse coding framework can enhance empirically developed forecasting strategies.

**Setting**  Let us consider the following forecasting problem, which resembles the online learning game introduced at the beginning of this paper. The difference is that, here, we further assume access to a black box forecaster $\mathcal{A}$. In each (the $t$-th) round,

1. The black box forecaster $\mathcal{A}$ produces a prediction $a_t \in \mathbb{R}^d$ based on the observed history ($z_{1:t-1}$ and $l_{1:t-1}$).
2. After observing $a_t$, we make a prediction $x_t \in \mathbb{R}^d$.
3. The environment reveals a true value $z_t \in \mathbb{R}^d$ and a convex loss function $l_t : \mathbb{R}^d \to \mathbb{R}$. $l_t$ is $G$-Lipschitz with respect to $\|\cdot\|_2$, and $z_t$ is one of its minimizer satisfying $l_t(z_t) = 0$.

Our goal is to achieve low total loss $\sum_{t=1}^T l_t(x_t)$. Since trivially picking $x_t = a_t$ already achieves a total loss of $\sum_{t=1}^T l_t(a_t)$, our goal is to improve it in certain situations, by designing a more sophisticated prediction rule based on $a_t$.

**Intuition**  In the above setting, $\mathcal{A}$ can be *any* algorithm that predicts $z_{1:T}$ in a reasonable, but non-robust manner. Taking the weather forecasting for example, there are a few notable cases.

- $\mathcal{A}$ is a simulator of the governing meteorological equations, which uses the online observations $z_{1:t-1}$ as boundary conditions.
- $\mathcal{A}$ is an autoregressive model, which predicts a linear combination of the past observations. The coefficients are determined by statistical modeling.
- $\mathcal{A}$ is a large deep learning model trained on offline datasets (e.g., the weather history at geographically similar locations).

Even though such forecasters typically lack performance guarantees, their predictions can be used to construct time-varying Bayesian priors (see our discussion in the Introduction): given $a_t$, we will apply a *fine-tuning* adjustment $\delta_t$ to predict $x_t = a_t + \delta_t$. Intuitively, the total loss is low if $a_t$ is close to the true value $z_t$, i.e., when the prior is good.

**Reduction to unconstrained dynamic regret**  Concretely, if $x_t = a_t + \delta_t$, then due to convexity, for all subgradients $g_t \in \partial l_t(x_t)$ we have $l_t(x_t) - l_t(z_t) \leq \langle g_t, \delta_t \rangle - \langle g_t, z_t - a_t \rangle$. The RHS is the instantaneous regret of $\delta_t$ in an OLO problem with loss gradient $g_t$ and comparator $z_t - a_t$. Applying our unconstrained dynamic OLO algorithm, the total loss in forecasting can be bounded as

$$\sum_{t=1}^T l_t(x_t) \leq \mathrm{Regret}_T(z_{1:T} - a_{1:T}).$$

That is, the total loss bound adapts to the complexity of the *error sequence* $z_{1:T} - a_{1:T}$ (of the given black box forecaster). This contains $a_{1:T} = 0$ as a special case, where no side information is assumed.

Let us compare this bound to the baseline $\sum_{t=1}^T l_t(a_t)$, which corresponds to trivially picking $x_t = a_t$.

- If $z_{1:T} = a_{1:T}$, i.e., the black box $\mathcal{A}$ is perfect, then the baseline loss is $\sum_{t=1}^T l_t(a_t) = 0$. In this case, due to Theorem 4, our general sparse coding framework guarantees $\sum_{t=1}^T l_t(x_t) \leq \varepsilon G$, where $\varepsilon > 0$ is an arbitrary hyperparameter. That is, our algorithm is worse than the baseline by at most a constant.

---

[15]Code is available at https://github.com/zhiyuzz/NeurIPS2023-Sparse-Coding.

- If $z_{1:T} \neq a_{1:T}$, then in general, the baseline loss $\sum_{t=1}^{T} l_t(a_t)$ is linear in $T$. In contrast, our algorithm could guarantee a sublinear $\text{Regret}_T(z_{1:T} - a_{1:T})$, thus also a sublinear total loss, when the error sequence $z_{1:T} - a_{1:T}$ is structurally simple (e.g., sparse under a transform, or low path length) with respect to our prior knowledge.

In summary, the idea is that by sacrificing at most a constant loss when $\mathcal{A}$ is perfect ($z_{1:T} = a_{1:T}$), we could robustify $\mathcal{A}$ against certain structured unseen environments, improving the linear total loss to a sublinear rate.

**Importance of unconstrained domain**    The above application critically relies on the ability of our algorithm to handle unconstrained domains. To demonstrate this, suppose we instead use the bounded domain algorithm from [ZLZ18] to pick the fine-tuning adjustment $\delta_t$. Then, the above analysis only holds if an upper bound $D$ of the maximum error $\max_t \|z_t - a_t\|_2$ is known a priori – this is a stringent requirement in practice. Furthermore, when $z_{1:T} = a_{1:T}$, such an alternative approach only guarantees $\sum_{t=1}^{T} l_t(x_t) \leq \tilde{O}(D\sqrt{T})$, which is considerably worse than the baseline 0. In other words, the alternative fine-tuning strategy could ruin the black box forecaster $\mathcal{A}$, when the latter performs well.

In the rest of this section, we present experiments for this time series application. Appendix E.1 demonstrates the power law phenomenon, which shows that both the time series $z_{1:T}$ and the error sequence $z_{1:T} - a_{1:T}$ could exhibit exploitable structures. This implies good performance guarantees using our theoretical framework. Appendix E.2 goes one step further by actually testing the fine-tuning performance of our algorithm.

### E.1    Power law phenomenon

This subsection further verifies the power law phenomenon discussed in Section 2.2, with both wavelet and Fourier dictionaries. The goal is to present concrete examples where signal structures can be exploited by our framework, generating more interpretable, sublinear regret bounds.

**Wavelet dictionary**    We first verify the power law on the Haar wavelet dictionary. Intuitively it is suitable when the dynamics of the environment exhibits switching behavior. To this end, consider the following stochastic time series model

$$z_t = z_{t-1}\beta_t + \zeta_t, \tag{17}$$

where $\{\beta_t\}$ and $\{\zeta_t\}$ are iid random variables satisfying $\zeta_t \sim \text{Uniform}(-q, q)$ and

$$\beta_t = \begin{cases} -1, & \text{w.p.} \quad p, \\ 1, & \text{w.p.} \quad 1-p. \end{cases}$$

Picking $T = 2^{15} = 32768$, $p = 0.0005$ and $q = 0.005$, we generate four sample paths of $z_{1:T}$ using four *arbitrary* random seeds (2020, 2021, 2022 and 2023), and the obtained time domain signals are plotted in the first row of Figure 2. As the switching probability $p$ is chosen to be low enough, all the sample paths exhibit a small amount of sharp switches, corrupted by the noise term $\zeta_t$. According to our intuition from signal processing, the Haar wavelet transform of these signals is sparse.

Now let us verify this intuition. We take the Haar wavelet transform of these signals, sort the transform domain coefficients and plot the results on log-log scales – these are shown as the solid blue lines in the second row of Figure 2. Using the largest 100 transform domain coefficients on each plot, we fit a liner model using least square, which is shown as the dashed orange line. The slope of each line is $-\alpha$, where $\alpha$ is displayed in the legend. It can be seen that for all four sample paths, the fitted $\alpha$ is within $(0.5, 1)$, thus justifying the power law phenomenon [Pri21]. Given $\alpha$, the regret of our Haar wavelet algorithm is $\tilde{O}(T^{1.5-\alpha})$, as shown in Section 2.2.

As for the implication in time series forecasting, let us consider forecasting $z_{1:T}$ with $a_{1:T} = 0$, i.e., without the external forecaster $\mathcal{A}$. Given the power law, the total forecasting loss of our fine-tuning approach is $\sum_{t=1}^{T} l_t(x_t) \leq \tilde{O}(T^{1.5-\alpha})$.

We also remark that although only four sample paths are demonstrated, we observe the power law phenomenon on all random seeds we tried in the experiment.

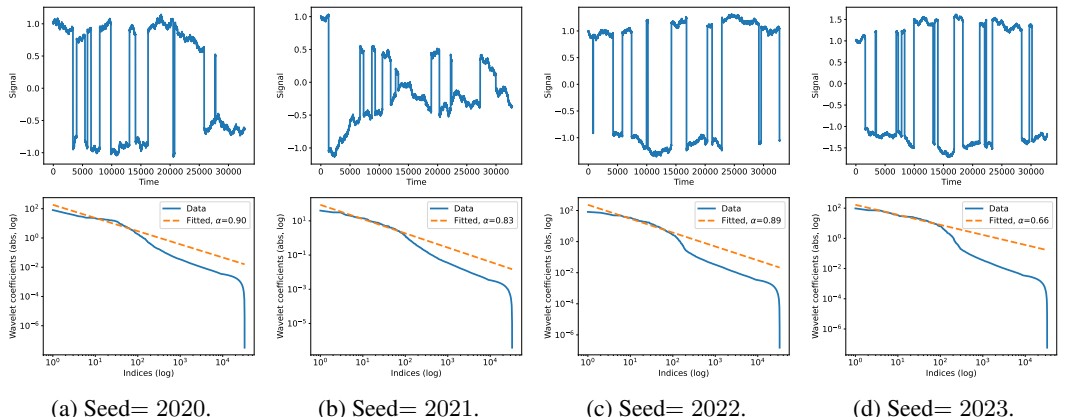

(a) Seed= 2020.  (b) Seed= 2021.  (c) Seed= 2022.  (d) Seed= 2023.

Figure 2: Verifying the power law on the Haar Wavelet dictionary. First row: time domain signals. Second row: sorted transform domain coefficients on a log-log plot. The dashed orange line is the best linear fit on the log-log plot, using the largest 100 transform domain coefficients. From left to right: four arbitrary random seeds.

**Fourier dictionary**    Next, we verify the power law on the Fourier dictionary. Here we use the Jena weather forecasting dataset,[16] which records the weather data at a German city, Jena, every 10 minutes. We take the data from Jan 1st, 2010 till July 1st, 2022, consisting of $T = 656956$ time steps. Two different modalities, namely the temperature and the humidity, are considered. The actual temperature and humidity sequences are plotted in Figure 3.

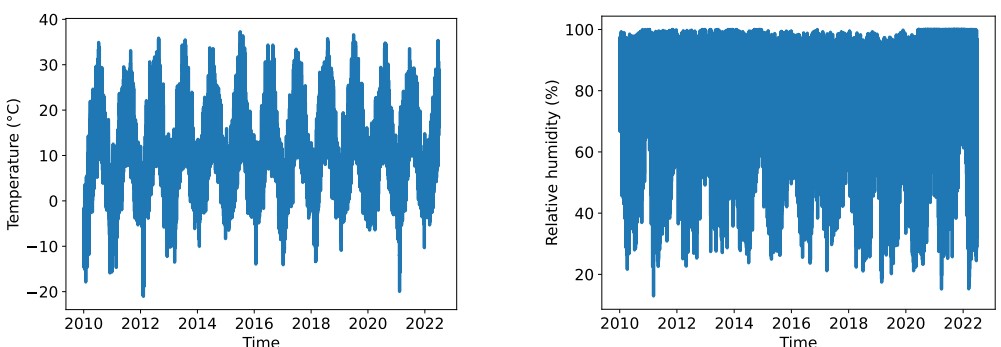

Figure 3: Time domain behavior of the weather data.

For the sequence of temperature $z_{1:T}$, we perform its *Discrete Fourier Transform* (DFT), which returns $T$ complex number as the frequency domain coefficients. We discard the second half of the coefficients due to symmetry, since the input of the transform is real. For the remaining coefficients, we take their absolute values, sort them and plot the result on a log-log plot. Similar to the wavelet experiment, we also fit a linear model using the largest 100 transform domain coefficients. These are shown as Figure 4 (Left), which exhibit the power law phenomenon.

Furthermore, we perform the same procedure on the temperature difference sequence $\{z_{t+1} - z_t\}$, where the $t$-th entry is the change of temperature from the $t$-th time step to the $t + 1$-th time step. The result is shown as Figure 4 (Right). Although the tail is heavier, we can still observe similar power-law phenomenon for large transform domain coefficients.

Now, let us discuss again the implication of the observed power law in time series forecasting. First, consider forecasting $z_{1:T}$ without $\mathcal{A}$. Given the power law of $z_{1:T}$ itself, the Fourier version of our forecaster guarantees sublinear total loss. Next, consider forecasting $z_{1:T}$ with $\mathcal{A}$ being the *zeroth-order hold* forecaster, i.e, $a_t = z_{t-1}$. The power law of the difference sequence $\{z_{t+1} - z_t\}$ implies good forecasting performance of our framework in this context.

---

[16]Available at https://www.bgc-jena.mpg.de/wetter/.

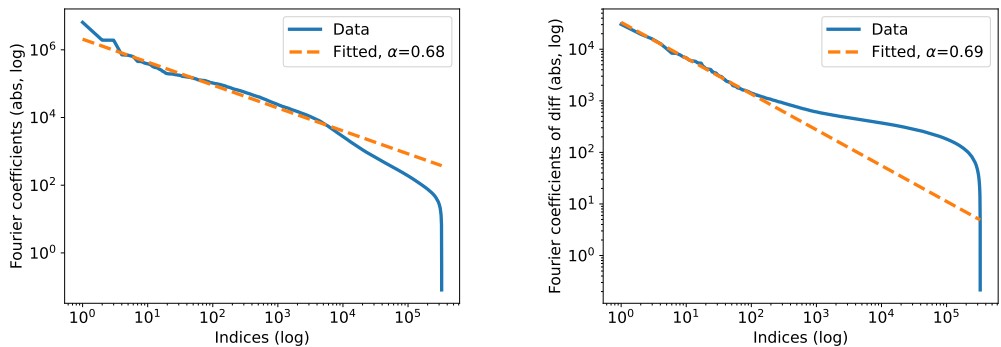

Figure 4: Verifying the power law on the Fourier dictionary. Left: the DFT of the temperature sequence. Right: the DFT of the temperature difference sequence.

Parallel results on the humidity sequence are reported in Figure 5, with a similar qualitative behavior. It illustrates the prevalence of the power law across different types of the data.

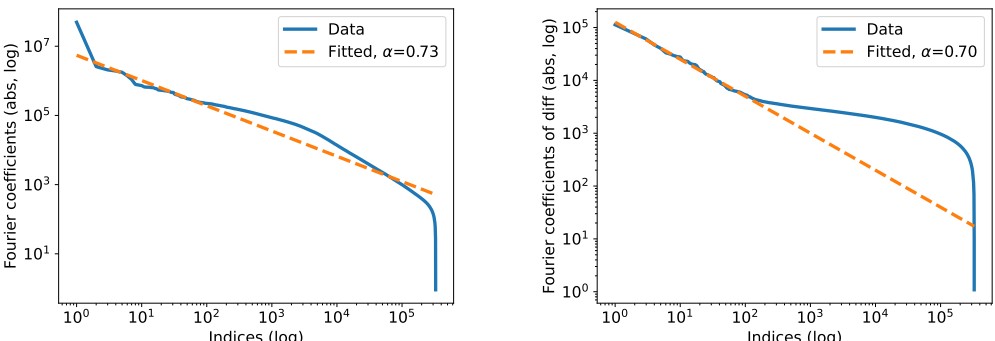

Figure 5: Verifying the power law on the Fourier dictionary. Left: the DFT of the humidity sequence. Right: the DFT of the humidity difference sequence.

### E.2 Fine-tuning forecaster

Finally, we test the performance of our forecasting framework on the synthetic switching data and the actual temperature sequence. For the first case, our framework is equipped with the Haar wavelet dictionary. The Fourier dictionary is adopted in the second case. In both cases, we compare our algorithm against the baseline from [JC22]. More specifically, we take our online learning algorithm (Algorithm 2) and the algorithm from [JC22], plug them both into the time series forecasting workflow introduced at the beginning of this section, and then compare their total forecasting loss.

Concretely, let us start from the wavelet dictionary.

**Wavelet dictionary** In this case, consider the setting without the external forecaster $\mathcal{A}$. We run both online learning algorithms (our Algorithm 2 and the baseline [JC22, Algorithm 2]), and use their outputs directly as the time series predictions. Our Algorithm 2 is equipped with the Haar wavelet dictionary defined in Section 3.1. The configurations of the time series model are the same as the previous subsection, with $T = 2^{15} = 32768$, $p = 0.0005$ and $q = 0.005$. The loss functions $l_t$ are the absolute loss.

Both algorithms require a confidence hyperparameter $\varepsilon$, and we set it to 1. Since the time series data Eq.(17) is random, we run both algorithms on 10 random seeds, and calculate their total loss. Our algorithm achieves a total loss of 44048, which is considerably lower than the baseline's total loss 62465. This is consistent with the theoretical results developed so far.

**Fourier dictionary**    Next, we turn to the task of temperature forecasting. The data is reported in the previous subsection. We take its first $T = 50000$ entries, and assign it to the true time series $z_{1:T}$; the loss functions $l_t$ are the absolute loss. The black box forecaster $\mathcal{A}$ is assigned to the zeroth-order hold forecaster, i.e., $a_t = z_{t-1}$.

For our framework, we need to specify the dictionary. Although using the entire DFT matrix could lead to low regret guarantees (as demonstrated by the power law), this is computationally challenging. Instead, we exploit the fact that the weather is naturally periodic, with the period of one day. Picking the base frequency $\omega$ accordingly, we define features indexed by $k$ (the harmonic order) as

$$h_{t,2k-1} = \cos(k\omega t),$$

$$h_{t,2k} = \sin(k\omega t).$$

By specifying the maximum order $K$, we obtain $2K$ features $\{h_{t,2k-1}, h_{t,2k}\}_{k \in [1:K]}$ from this construction. An all-one feature is further added, making the dictionary size $N = 2K + 1$.

Again, we set the confidence hyperparameter $\varepsilon = 1$ for our algorithm. The total loss as a function of the dictionary size $N$ is plotted in Figure 6. Notably, the case of $N = 0$ is equivalent to trivially following the advice of the given forecaster $\mathcal{A}$: $x_t = a_t = z_{t-1}$. It can be seen that our fine-tuning framework ($N > 0$) actually results in better performance, due to exploiting the structures in the error sequence $z_{1:T} - a_{1:T}$.

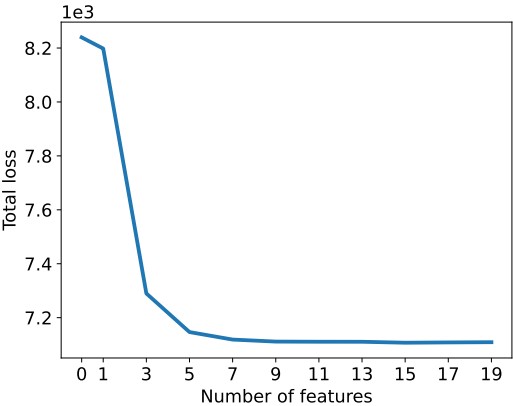

Figure 6: Testing our algorithm for temperature forecasting.

We also test the fine-tuning performance of the algorithm from [JC22]. Same as the above, we set $\varepsilon = 1$. The total loss achieved by this alternative algorithm is 8238, which is around the same as $\mathcal{A}$ itself, and significantly higher than the total loss of our algorithm with moderate amount of features ($N > 5$). This fits the intuition from this paper: the environment contains persistent dynamics, which the algorithm from [JC22] cannot handle.

# F    Additional discussion

**MRA in online learning**    On a broader scope, wavelets embody the idea of *Multi-Resolution Analysis* (MRA), which is reminiscent of the classical *geometric covering* (GC) construction in adaptive online learning [DGSS15]. Such a construction starts from a class of *GC time intervals*, which are equivalent to the support of Haar wavelet features. On each GC interval, a static online learning algorithm is defined (corresponding to using an all-one feature, c.f., Section 2.2); and then, the outputs of these "local" algorithms are aggregated by a *sleeping expert* algorithm on top [LS15, JOWW17]. Algorithmically, our innovation is introducing sign changes in the features, accompanied by a different, additive way to aggregate base algorithms. For tackling nonstationarity, both approaches have their own strengths: the GC construction can produce *strongly adaptive* guarantees on subintervals of the time horizon, while our algorithm does not need a bounded domain. Their possible connections are intriguing.

**Lipschitz vs strongly convex losses**   We also comment on the choice of loss functions in unconstrained dynamic OCO. Besides the Lipschitz assumption we impose, a fruitful line of works by Baby and Wang [BW19, BW20, BW21, BZW21, BW22] considered an alternative setting with strong convexity, motivated by the prevalence of the square loss in statistics. Their focus is primarily on bounded domains, as [BW19] showed that evaluated under the square loss, a lower bound for the unconstrained dynamic regret is $\Omega(P^2)$. A sublinear regret bound here requires $P = o(\sqrt{T})$, rather than $P = o(T)$ with Lipschitz losses – that is, the environment is required to be "more static" than the typical requirement in the Lipschitz setting.

Essentially, such a behavior is due to the large penalty that the square loss imposes on outliers. An adversary in online learning can deliberately pick the loss functions such that some of the player's predictions are large outliers with "huge" (square) losses, while the offline optimal comparator sequence suffers zero losses. Using the Lipschitz losses instead may offer an advantage on unbounded domains, due to being more tolerant to these outliers. Furthermore, Lipschitz losses do not necessarily have minimizers – this is useful for *decision* problems (as opposed to *estimation*), where a ground truth may not exist.[17]

**Future work**   For future works, several interesting questions could stem from this paper. For example,

- Our regret bound is stated against individual comparator sequences. One could investigate the implication of this result in stochastic environments, where the comparator statistics may take more concrete forms.
- Besides the sparsity and the energy studied in this paper, an interesting open problem is investigating alternative complexity measures of the comparator, possibly drawing connections to statistical learning theory.
- Our framework builds on pre-defined dictionary inputs. The quantitative benefit of using a data-dependent dictionary is unclear.
- Beyond wavelets, one may investigate the combination of the sparse coding framework with other function approximators, such as neural networks.

---

[17]An example is financial investment without budget constraints: doubling the invested amount also doubles the return.

