# OpenReview forum: "Unconstrained Dynamic Regret via Sparse Coding"
_NeurIPS.cc/2023/Conference — NeurIPS 2023 poster_

### Official Review · Reviewer_4s3X · 2023-06-30

**Soundness:** 2 fair
**Presentation:** 2 fair
**Contribution:** 3 good
**Rating:** 6
**Confidence:** 2

**Summary:**

This paper studies adaptive online convex optimization, with focus on adapting to unbounded domain and arbitrary time-varying comparator sequence. Different from previous studies which mostly consider the path-length as the prior which appears in the dynamic regret bound, this paper aims to enlarge the range of priors. For a given dictionary of orthogonal feature vectors, it is shown that an $\tilde{O}(\sqrt{E\cdot \text{Sparsity}_H})$ regret bound can be achieved, which not only recovers previous results on path-length, but also allows more flexibility. This paper also strictly improves the result of JC22 by replacing comparator-dependent terms in the regret bound by strictly smaller ones.

**Strengths:**

This paper makes solid contributions to adaptive online convex optimization. It extends the scope of comparator adaptivity from the traditional path-length, to other priors on the comparator via a novel sparse coding framework. The achieved bounds involve more refined dependence on the comparator, and are never worse than previous results. In particular, the special case of the wavelet dictionary improves the SotA result in JC22.

**Weaknesses:**

The main weakness is the unclear presentation of technical results, which are hard to fully understand and interpret. Though the claimed results seem promising, I have to vote for rejection for now because I can't verify the correctness without a better understanding of the results. Here are some confusions I met.

For the size 1 dictionary case, isn't the assumption too strong which seems to trivialize Lemma 2.1? In line 211, the comparator is assumed to lie in the span of a single vector $h_{1:T}$, which is just a scaling of the vector $h_{1:T}$. Since the dictionary is fixed and revealed to the player, the range of potential comparators seems very restricted (the degree of freedom is just one through $\hat{u}$).

The same issue carries over to the main result Theorem 1, which assumes the signal $z^{(n)}$ lies in the span of a single vector. Even if the reconstruction error creates certain flexibility, to achieve vanishing regret one still needs to guarantee the comparator is very close to $\sum_{n=1}^N z^{(n)}$, which seems under-expressive: for a given dictionary, the overall degree of freedom is just $n$, while for arbitrary time-varying comparators the degree of freedom is $dT$.

Two examples are presented right after the main theorem. The case of static regret is well understood. However, I feel the case of orthogonal dictionary requires more technical details, at least to readers unfamiliar with signal processing. For example, what exactly does the "orthogonal" means here? (which vector is orthogonal to which?) It would be great if you can provide an intuitive example here to show the power of your method over previous results.

I'm also concerned with how to choose the dictionary. It seems a good choice of dictionary requires prior knowledge. It's not clear to me if one can adapt to a class of dictionaries.

**Questions:**

Can you provide more technical details to address my questions discussed above? In particular, can you provide 1: an example of dictionary that recovers the traditional path-length case, and 2: an example of dictionary that corresponds to some other prior different from path-length?

Can you provide details on using the Fourier dictionary to tackle periodic environments? It looks interesting to me because the naive idea of setting $H_t=(-1)^t I_d$ won't work.

**Limitations:**

The presentation of technical results is not very clear.

---

> ### Author Rebuttal · Authors · 2023-08-07
>
> Thanks for your comments! We hope the following clarifications can answer your questions.
>
> - Interpretation of the generic framework
>
> Our generic framework aggregates a collection of single-feature learners. Roughly speaking, each of these single-feature learner is in charge of a fixed direction in the sequence space $\mathbb{R}^{dT}$. Aggregating them means assembling these fixed directions into a subspace in $\mathbb{R}^{dT}$, which the comparator sequence $u_{1:T}$ belongs to.
>
> To recover static online learning, we need $N=d$ feature vectors. The dynamic setting is more challenging: in Theorem 1, if we want to completely eliminate the reconstruction error term $z^{0}$, then we have to use $dT$ feature vectors. As mentioned in your comment, this means setting $N=dT$.
>
> You may wonder if this procedure is computationally efficient, since there are now $dT$ simple learners to aggregate. This is the reason why we focus on the wavelet dictionary in the second part of the paper: among the $O(T)$ simple learners, only $O(\log T)$ of them are active in any given round, therefore computationally their aggregation is not so expensive.
>
> - Meaning of "orthogonal" and example
>
> "Orthogonal dictionary" in this paper means that we consider a dictionary matrix $\mathcal{H}\in\mathbb{R}^{dT\times N}$ whose columns are orthogonal. An example is the Haar wavelet dictionary, e.g., Eq.(8).
>
> To recover (and improve) traditional path-length bounds, we can use the Haar wavelet dictionary.
>
> For a different inductive bias, the Discrete Fourier Transform (DFT) matrix is useful for periodic environments. Besides, the Daubechies wavelet family is a generalization of the Haar wavelet suitable for piecewise smooth environments.
>
> - How to choose the dictionary, and adaptivity
>
> The choice of the dictionary relies on (Bayesian) prior knowledge of the environment, which is a key idea of adaptive online learning, and machine learning in general. For *any* dynamic OCO algorithm, if the environment behaves very differently from the inductive bias of the algorithm (e.g., the environment is periodic, but the algorithm guarantees a path-length-based bound), then its performance cannot be good. Our framework is not an exception.
>
> Meanwhile, it is indeed possible to "adapt" to the best dictionary in a certain category: given a collection of dictionaries, we can perform almost as if we know beforehand which dictionary is the best one. This is explained in our line 251 to 257. The idea is to simply combine all the dictionaries into a mega-dictionary, and run our framework verbatim. The dictionary-selection property follows from some nice behaviors of static comparator-adaptive online learning. Achieving this task without using the mega-dictionary is an interesting open question.
>
> - Fourier dictionary
>
> For periodic environments, the most natural idea is to use the Discrete Fourier Transform (DFT) matrix as the dictionary. The limitation is that the DFT matrix is dense itself, therefore computationally, this approach needs to aggregate $dT$ simple learners per round, which is computationally challenging.
>
> In practice, a more appealing approach is to use a smaller dictionary defined from a base frequency and its low order harmonics. Specifically in the one-dimensional setting, given a base frequency $\omega$ and a maximum order $K$, we define two features for all $k\leq K$: the first has per-round component $h_t=\cos(k\omega t)\in\mathbb{R}$, and similarly, the second is $h_t=\sin(k\omega t)$. The base frequency $\omega$ is often determined by the natural periodicity of the environment, e.g., the weather and the traffic flow are roughly daily periodic. Details and supporting experiments are presented on the last page of the appendix. In the weather forecasting experiment, using a moderate amount of features ($K>3$) suffices, which quite significantly outperforms the baseline from [JC22].

---

> > ### Comment · Reviewer_4s3X · 2023-08-11
> > **Reply to Authors**
> >
> > Thank you for your detailed reply.
> >
> > In my opinion, the general result seems less interesting than its special cases: to recover the general dynamic setting, the algorithm requires $dT$ feature vectors, which is unpractical. On the other hand, the Haar wavelet version does provide some solid improvements over previous results.
> >
> > Though this work has certain limitations currently, the signal-processing perspective is novel and promising, which may open a new avenue for future research. I have updated my score accordingly.

---

> > > ### Author Response · Authors · 2023-08-11
> > >
> > > Thanks for carefully evaluating our work!

---

### Official Review · Reviewer_pdGo · 2023-07-05

**Soundness:** 3 good
**Presentation:** 3 good
**Contribution:** 2 fair
**Rating:** 4
**Confidence:** 4

**Summary:**

The problem that this paper tries to tackle is the unconstrained online convex optimization with dynamic regret. Previous works usually assume that the comparator sequence is arbitrary and maybe time-varying with some fixed form of comparator measurement in the final dynamic regret bound. For this paper, it proposes a new way of the comparator measurement by first using a pre-defined subspace (by the users) to characterize the comparator sequence and then upper bound the dynamic regret in terms of the pre-defined subspace complexity as well as the reconstruction error between the subspace and the comparator sequence. It shows that for the almost static environment, the proposed algorithm + wavelet constructed subspace could have better regret than the existing works' in (maybe) the constant part.

**Strengths:**

1) It provides the researchers a new perspective of measuring the comparator sequence when upper bounding the dynamic regret, which is interesting and useful in real applications.
2) The specific wavelet based subspace version could result in a regret better than the existing works' with (maybe) smaller constant part.

**Weaknesses:**

1) The proposed algorithm framework only moves the dynamic regret from previous works in the constant value part. And in order to compare the actual improvement in terms of the constant part, the author/s need to provide the complete regret result comparison instead of an order O() based one.
2) The paper result depends heavily on the existing work [MK20], and the subspace based comparator sequence measurement is the only part that makes it interesting, although previous works like [HW15, ZLZ18] has already shown such an idea as pointed out by the author/s.
3) The author/s explained the motivation of tackling unconstrained dynamic setting rather than the constrained one. But for the finite range argument in the paper, more often than not, the finite range usually comes from the requirement of the model output and not just some heuristic estimation. I think it's better to also provide some examples to demonstrate the motivation of this paper.

**Questions:**

1) The proposed algorithm requires the user to provide the subspace, which is very critical for the final dynamic regret result. If the user provided subspace is not good enough, does that mean the resulted algorithm will have bad performance? If so, how to guarantee the performance?
2) Since the proposed algorithm depends on the Algorithm 3 a lot, do you think it may make more sense to move Algorithm 3 from the Appendix to the main paper?

---

> ### Author Rebuttal · Authors · 2023-08-07
>
> Thanks for your feedback!
>
> - Improvement over [JC22].
>
> We would like to respectfully clarify that compared to [JC22], our bound depends on a tighter complexity measure of the comparator sequence $u_{1:T}$. Such an improvement is considerably more substantial than improving the multiplicative constant of the conventional minimax regret bounds. In fact, multiplicative constants and logarithmic factors are omitted in our analysis. Example 1 and 2 demonstrate improvements on the *exponent* of $T$, rather than multiplying factors.
>
> - Dependence on [MK20]
>
> Our framework is actually very general: any static comparator adaptive OCO algorithm can be applied to replace [MK20], such as [MO14, OP16, CO18]. We only adopt [MK20] as an illustrative example.
>
> - Difference with [HW15, ZLZ18]
>
> As discussed in the paper, our framework is fundamentally different from [HW15, ZLZ18], as we use *linear combinations* of the side information, rather than *convex combinations*, to approximate the comparator sequence. This is the key reason behind its success, as it leads to natural connections to linear transforms, one of the most fundamental ideas of signal processing.
>
> Besides, we would like to argue that despite bearing a natural idea, the analysis in this paper is a nontrivial one, and the strength of the wavelet result is quite surprising in our opinion.
>
> - Bounded domain setting
>
> If we are given a bounded domain, then there is a useful projection technique [Section 4, Cut20] that converts an unconstrained algorithm to a constrained one, without changing its regret bound. Therefore, our unconstrained dynamic regret bound also improves the standard bounded domain dynamic regret bound from [ZLZ18], modulo logarithmic factors.
>
> Moreover, Appendix E discusses an application in fine-tuning time-series forecasters, which quite naturally motivates our unconstrained setting (due to its enhanced adaptivity). The relevant discussion is line 903 to 915.
>
> - Dependence on the quality of the dictionary
>
> Indeed, the performance of our generic framework depends on the quality of the dictionary. The rationale is a classical and natural one: without a good inductive bias from the dictionary, we cannot compete with a benchmark that "knows the future''. In a consistent manner, all the existing dynamic regret bounds are trivially $O(T)$ in their worst case.
>
> With this, the main strength of our framework is its versatility: it is strictly more general than existing approaches without dictionaries. One could always pick the Haar wavelet dictionary, which leads to better quantitative bounds than the baselines, in almost static environments.

---

> > ### Comment · Reviewer_pdGo · 2023-08-18
> >
> > I appreciate the rebuttal. I agreed that the regret bound depends on a tighter complexity measure of the comparator sequence. But the Eq.(7) on the sparsity claim is a bit exaggerated. Although it does have the sparsity indicated there, if taking the E term together, it's actually a reformulation of the numerator term from sparsity. Although I agree it's still tighter than previous works results, its improvement is not that significant. Since my novelty concerns are not well addressed, I will keep my score unchanged.

---

### Official Review · Reviewer_45WG · 2023-07-06

**Soundness:** 4 excellent
**Presentation:** 4 excellent
**Contribution:** 4 excellent
**Rating:** 7
**Confidence:** 4

**Summary:**

In this paper, the authors examine the dynamic regret of Online Convex Optimization (OCO) within the context of unbounded comparator sequences. To address this issue, they introduce a novel framework of sparse dictionary coding for online optimization. Following this, the authors provide theoretical proof for the regret bounds applicable to different types of dictionary matrices - the general dictionary matrix, orthogonal dictionary matrix, and the Haar wavelet dictionary matrix. Of notable mention is the result pertaining to the Haar wavelet dictionary matrix, where the authors establish a regret bound that surpasses the current state-of-the-art.

**Strengths:**

- Originality & Significance: The introduction of the sparse coding framework for Online Convex Optimization (OCO), complemented with the corresponding proof, is highly innovative. Furthermore, considering the dynamic regret of OCO in the unbounded domain is very crucial. The result that the authors obtained is related to the comparator average, first-order variability, and second-order variability, rather than the path length. This indicates that the authors have achieved smaller dynamic regret under more adaptive conditions, thereby enriching optimization theory within the community.
- Quality & Clarity: The work is clear, well written, and technically sound.

**Weaknesses:**

- Comparing the proposed method with the meta-expert optimistic online gradient descent method as described in [1] could be beneficial. Particularly in Examples 1 and 2, it appears that the meta-expert optimistic online gradient descent method can also achieve a regret of $\mathcal O(\sqrt T)$, considering the path-length is actually $\mathcal O(\sqrt T)$. This comparison might offer a broader perspective.

- I'm concerned about the computational complexity of $\mathcal O(d\log T)$ for high-dimensional problems, especially when compared to the $\mathcal O(\log T)$ complexity of both the meta-expert OGD and the meta-expert optimistic OGD methods. If possible, an in-depth discussion on this issue would be helpful for readers.

- Additionally, the main text seems to miss Algorithms 3-5. This omission may cause a minor confusion for readers during their first read. Rectifying this could help in enhancing the flow and clarity of the paper.

Ref: [1] P Zhao et.al., Adaptivity and Non-stationarity: Problem-dependent Dynamic Regret for Online Convex Optimization.

**Questions:**

See above discussions.

---

> ### Author Rebuttal · Authors · 2023-08-07
>
> Thanks for your feedback and your support of our paper!
>
> - Related work [1]
>
> Thanks for bringing it to our attention. Both [1] and our paper study how to achieve more adaptivity in dynamic online learning, but they take different directions. For example, [1] has an additional smoothness assumption, therefore the quantitative results are not directly comparable to ours. We'll discuss this related work in the camera ready version.
>
> - Computational complexity of meta-expert OGD
>
> It's possible that we missed its latest improvement: to our knowledge, the meta-expert OGD baseline [Ader, ZLZ18] also runs in $O(d\log T)$ time per round right? There are $O(\log T)$ base algorithms running in parallel, and each of them is OGD in $\mathbb{R}^d$ which runs in $O(d)$ time per round.
>
> - Finally, thanks for the suggestions on the organization of this paper.

---

> > ### Comment · Reviewer_45WG · 2023-08-21
> >
> > I'd like to express my gratitude for the author's clarifications and comments. Indeed, the meta-expert OGD runs in $\mathcal O(d\log T)$ time per round. I acknowledge my mistake and appreciate the correction made by the author.

---

### Official Review · Reviewer_qkG5 · 2023-07-06

**Soundness:** 3 good
**Presentation:** 3 good
**Contribution:** 3 good
**Rating:** 7
**Confidence:** 3

**Summary:**

This paper studies the universal dynamic regret minimization problem with the unconstraint decision domain. The authors proposed the sparsing coding framework, which converts the dynamic regret minimization problem in the time domain into a static regret minimization problem in the transfer domain. The comparator-adaptive static regret bound in the transfer domain implies a dynamic regret bound in the time domain. Specifically, by choosing the dictionary as the Haar wavelet base, this paper achieves improved dynamic regret bound with better dependence on the range of the comparator sequence. Several concrete examples are provided to illustrate the superiority of the proposed bound.

**Strengths:**

+ This paper provides a novel framework to obtain universal dynamic regret bound via sparsing coding. The conversion from comparator-adaptive static regret bound to dynamic regret bound in the time domain is interesting to me.
+ This paper achieves improved universal dynamic regret bound with the new framework, which is strictly better than the existing results. Several examples are provided to illustrate the advantages of the proposed bound.
+ This paper is well-written and provides a sufficient discussion of the related literature.

**Weaknesses:**

I do not find a major weakness in the paper, but there are still some minor comments:

- about the general formulation (Eq. (5)): it would be nice to mention that the dynamic regret bound (Eq.(5)) is not universal dynamic regret bound in general as it only holds for the comparator $u_{1:T}\in\mbox{span}(h_{1:T})$. The universal dynamic regret bound can only be achieved with an appropriate choice of the dictionary.

- about the examples: this paper has listed several examples with the specific choice of the comparator sequence to show the superiority of the proposed bound. However, since the main focus of this paper is the universal dynamic regret, it is unclear in which situation the listed comparator sequence is an appropriate benchmark that can minimize the right-hand side of Eq (2). I suggest the authors provide more concrete examples to illustrate the advantage of the proposed bound with certain specific loss functions.


**Questions:**

- could you provide more concrete examples to illustrate the superiority of the proposed bound? (please refer to the second point of the weakness for more details)
- is it possible to achieve the $\tilde{O}\left(\Vert \bar{u}\Vert\sqrt{T}+\sqrt{P\bar{S}}\right)$ bound without the knowledge of $T$. Could you highlight what is the main difference to obtain such a bound?

**Limitations:**

One of the main limitations of the paper is that the tightness of the proposed bound is still unclear. The authors discuss this issue at the end of the paper.

---

> ### Author Rebuttal · Authors · 2023-08-06
>
> Thanks for your comments and your support of our paper!
>
> - Clarification on Eq.(5).
>
> Thanks for the suggestion, we will add the remark that the bound holds for $u_{1:T}$ in the span of the feature vectors.
>
> - Example of the loss functions.
>
> The question is on whether there exist loss functions $l_{1:T}$ such that the comparators $u_{1:T}$ from Example 1 and 2 are actually good comparators with low cumulative loss. This is a great point, and let us use time series forecasting as an example (Appendix E).
>
> Consider the true time series $z_{1:T}$ being the sequences from our Example 1 and 2, and the loss functions in OCO are the absolute loss, $l_t(x)=|x-z_t|$. As the comparator sequence, the true time series $z_{1:T}$ suffers zero loss, therefore the total loss of our OCO algorithm is upper bounded by its regret bound with respect to $z_{1:T}$. In this case, our improved regret bound over [JC22] translates to a smaller total loss bound.
>
> Essentially, this example shows that the *restricted dynamic regret bound* obtained from our universal bound improves the one obtained from [JC22]. Such an argument is a relaxation of the oracle inequality Eq.(2), and a natural next step is to directly characterize the infimum on the RHS there. To our knowledge, this is a less studied topic within comparator adaptive online learning, which could be a good direction for future works.
>
> - Anytime $\tilde O\left(||\bar u||_2\sqrt{T}+\sqrt{P\bar S}\right)$ bound
>
> This can indeed be achieved from our core fixed-$T$ result, which we realized after the submission. Details are the following. We will add this small improvement to the camera ready version.
>
> In our current proof, we have (line 801 and 802),
>
> $
> \mathrm{Regret}(u_{1:T})\leq \tilde O\left(\sum_{m=1}^{m^*}||\bar u_m||_2\sqrt{2^m}\right)+\tilde O\left(\sqrt{P \bar S}\right).
> $
>
> For the first sum on the RHS, same as our proof of Theorem 2 (line 722 to 727),
>
> $
> \sum_{m=1}^{m^*}||\bar u_m||_2\sqrt{2^m}\leq \tilde O\left(||\bar u||_2\sqrt{T}+\sqrt{\bar E}\right).
> $
>
> The remaining task is to show that $\sqrt{\bar E}$ is dominated by $\sqrt{P\bar S}$, thus can be combined into the later. Plugging in their definitions, it suffices to show that for all $t$, $||u_t-\bar u||_2\leq P$. This follows from
>
> $
> ||u_t-\bar u||_2 \leq T^{-1} \sum_i ||u_t-u_i||_2\leq \max_i ||u_t-u_i||_2,
> $
>
> and for all $i,t\in[1:T]$, $||u_t-u_i||_2\leq P$.

---

### Decision · Program_Chairs · 2023-09-21

**Decision:**

Accept (poster)

**Comment:**

This paper proposes an algorithmic framework for minimizing unconstrained dynamic regret. Unlike existing methods that depend on the path length, the dynamic regret of the proposed method depends on both the energy of the comparator sequence, and its sparsity with respect to a dictionary. In the special case where the dictionary is a Haar wavelet dictionary, this framework gives better dynamic regret bounds than previous state-of-the-art. During the discussion, the novelty of the framework and the use of the sparse coding perspective in online convex optimization is appreciated. Even though there is some concern around the practicality of the framework in the general case, the improvements in special cases is interesting and show the promise of this new approach. I recommend acceptance.